# Middle Ordovician astrochronology decouples asteroid breakup from glacially-induced biotic radiations

Jan Audun Rasmussen 🄳 [1,2,5], Nicolas Thibault 🄳 [3,5 ✉] & Christian Mac Ørum Rasmussen 🄳 [4,5]

Meso-Cenozoic evidence suggests links between changes in the expression of orbital changes and millennia-scale climatic- and biotic variations, but proof for such shifts in orbital cyclicity farther back in geological time is lacking. Here, we report a 469-million-year-old Palaeozoic energy transfer from precession to 405 kyr eccentricity cycles that coincides with the start of the Great Ordovician Biodiversification Event (GOBE). Based on an early Middle Ordovician astronomically calibrated cyclostratigraphic framework we find this orbital change to succeed the onset of icehouse conditions by 200,000 years, suggesting a climatic origin. Recently, this icehouse was postulated to be facilitated by extra-terrestrial dust associated with an asteroid breakup. Our timescale, however, shows the meteor bombardment to post-date the icehouse by 800,000 years, instead pausing the GOBE 600,000 years after its initiation. Resolving Milankovitch cyclicity in deep time thus suggests universal orbital control in modulating climate, and maybe even biodiversity accumulation, through geological time.

[1] Museum Mors, Skarrehagevej 8, DK-7900 Nykøbing Mors, Denmark. [2] Natural History Museum of Denmark, University of Copenhagen, Øster Voldgade 5–7, DK-1350 Copenhagen K, Denmark. [3] Department of Geosciences and Natural Resource Management, University of Copenhagen, Øster Voldgade 10, DK-1350 Copenhagen K, Denmark. [4] GLOBE Institute, University of Copenhagen, Øster Voldgade 5–7, DK-1350 Copenhagen K, Denmark. [5] These authors contributed equally: Jan Audun Rasmussen, Nicolas Thibault, Christian Mac Ørum Rasmussen. ✉email: nt@ign.ku.dk

The GOBE marked a sudden rise in early Palaeozoic biodiversity accumulation[1]. Leading up to the event was a gradual change in ecosystem engineering from detritus-feeding, mainly benthic, Cambrian faunas to more complex, mainly suspension-feeding faunas during the earliest Ordovician that were able to utilize the entire water column[2]. This change facilitated more efficient niche partitioning and more stable ecosystems that allowed for a higher degree of genus resilience[3].

By the Middle Ordovician, these mainly intrinsic ecosystem changes benefitted from a sudden shift to a colder climate that lowered ocean surface temperatures to present-day levels[4,5]. The resulting fast rise in species richness that characterizes the GOBE was likely the greatest increase in marine biodiversity of the entire Phanerozoic[6]. However, what instigated this sudden cooling is still poorly understood. Recently, the meteorite fallout associated with the breakup of the L-chondrite parent body (LCPB) that occurred some 468.1 million years ago was suggested to be the facilitating factor behind the cooling[7]. This catastrophic event is chiefly witnessed by Middle Ordovician fossil meteorite-bearing intervals that are prominent in lower Darriwilian rock successions of China and Baltoscandia[8]. The hypothesis brought forward[7] was that dust originating from the LCPB-disruption was delivered rapidly to Earth[9], instigating climatic deterioration that led to the GOBE. The timing of the asteroid breakup in space and the resultant meteorite fallout on Earth is now well-constrained by extra-terrestrial chromite, ³He-data[7], as well as by cosmic ray exposure ages that are tied to a high-precision Zircon U-Pb date of 467.5 ± 0.28 Ma[10] from the meteorite-bearing interval in southern Sweden. However, this evidence does not align with the hypothesis that onset of icehouse conditions during the Middle Ordovician correlates with the events in space.

To resolve this matter, we here extract 17 and 20 kyr precession and 405 kyr eccentricity components from well-preserved marl–limestone alternations in the Middle Ordovician Steinsodden section in the Moelv area, southern Norway (Figs. 1 and 2) that allow us to build a precise astrochronologic time scale that deciphers the fascinating chain of events that occurred during the early Middle Ordovician Darriwilian Age.

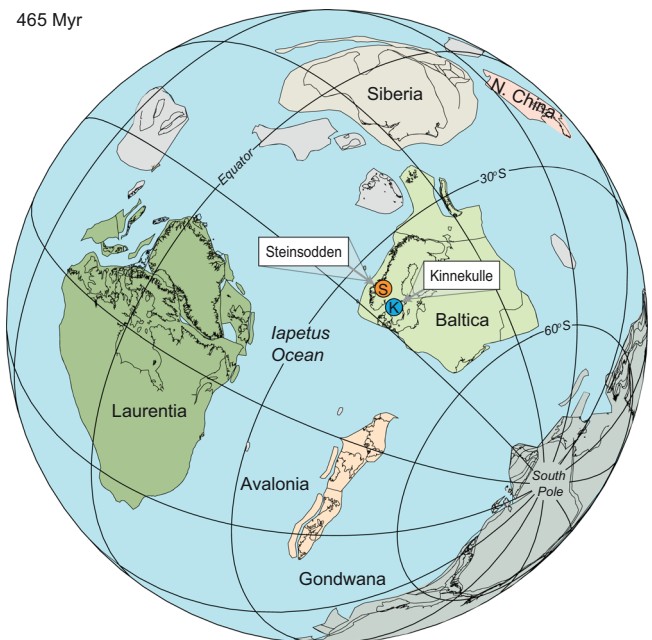

**Fig. 1 Middle Ordovician palaeogeographic configuration.** The Steinsodden section (S) and the Swedish locality Kinnekulle (K) highlighted. Map generated using the BugPlates software[61].

## Results

The 42 m thick Middle Ordovician (Dapingian–Darriwilian global stages) Stein Formation is exposed within the nature preservation area of Steinsodden, Moelv, Norway (60.906°N/10.696°E) (Figs. 2 and 3). This formation, which is part of the Lower Allochthon of the Norwegian and Swedish Caledonides, was deposited in an outer shelf palaeoenvironment[11,12]. The Stein Formation is characterized by regular alternations of dark grey argillaceous limestones and light grey beds of nearly pure limestone (Fig. 2). The 15–20 cm thick individual beds are fossiliferous and vary dominantly between mudstone and wackestone textures. The conodont biostratigraphy of this section is well-resolved[13]. Visible macrofossils occur only scattered, especially orthocerid cephalopods within the upper half part of the *L. variabilis* Zone, but fragments from mainly trilobites, brachiopods and crinoids are visible in thin sections (Supplementary Figs. 1–3).

**Detecting time cycles in the rock record**. Regular alternations of argillaceous limestone beds and beds of nearly pure limestone in outcrops have historically made strong cases for cyclostratigraphic analyses[14–16]. In the Stein Formation, which expresses these lithologies, we recognized six main lithofacies in the field (a–f in Table 1) and gave each a digital, lithologic rank value, where low values characterize softer and more clay-rich lithologies and high values represent pure limestones (Table 1).

The lithology ranks were assigned for each cm along the section, thus producing a discrete variable (Lithology rank, Supplementary Data 1) through a time-series of 3242 data points which constitute the framework for the cyclostratigraphic analyses.

**Orbital origin of marl–limestone alternations**. In the Baltoscandian epicontinental sea, the carbonate platform was subject to terrigenous influx from exposed hinterland. In the foreland basin of the Oslo Region, in the north-western part of Baltica (Fig. 1), periodic fluctuations in the terrigenous input were an important factor in the formation of rhythmic alternations of clay-rich and carbonate-rich beds. Terrigenous material was derived from (i) terrestrial areas towards the East and South during the Middle Ordovician, (ii) from the small Telemark Land area situated ~200–300 km SW of Steinsodden[17], and (iii) from evolving island arcs north of the palaeo-coastline[18]. In deeper subtidal, distal shelf environments like the Stein Formation, the carbonate supply was essentially carbonate mud derived from bio-erosion of platform carbonates because there was essentially no or very little in-situ pelagic carbonate production at that time[19]. Variations in carbonate content, and thus in lithofacies, could reflect either primary deposition, or an entirely diagenetic origin. Primary deposition of the Middle Ordovician rhythmic calcareous alternations would have been controlled via the interplay between the export of carbonate mud to the basin and the flux of terrigenous material controlled by runoff from the scattered terrestrial areas. In contrast, purely diagenetic rhythmic calcareous alternations have been explained by the self-organization of primarily homogeneous sediments during early diagenesis. Diagenetic redistribution of carbonate can potentially lead to the formation of purely diagenetic rhythmites through selective dissolution of aragonite and compaction in what became marl layers, and reprecipitation of calcite in what became limestones[20,21]. This diagenetic process is however generally restricted to shallow-water settings where the abundance of aragonitic shells is high. In contrast, the biomicritic limestones of the Steinsodden section show a dominance of skeletal debris of calcitic fossils dominated by trilobites and brachiopods, with rare gastropods[13].

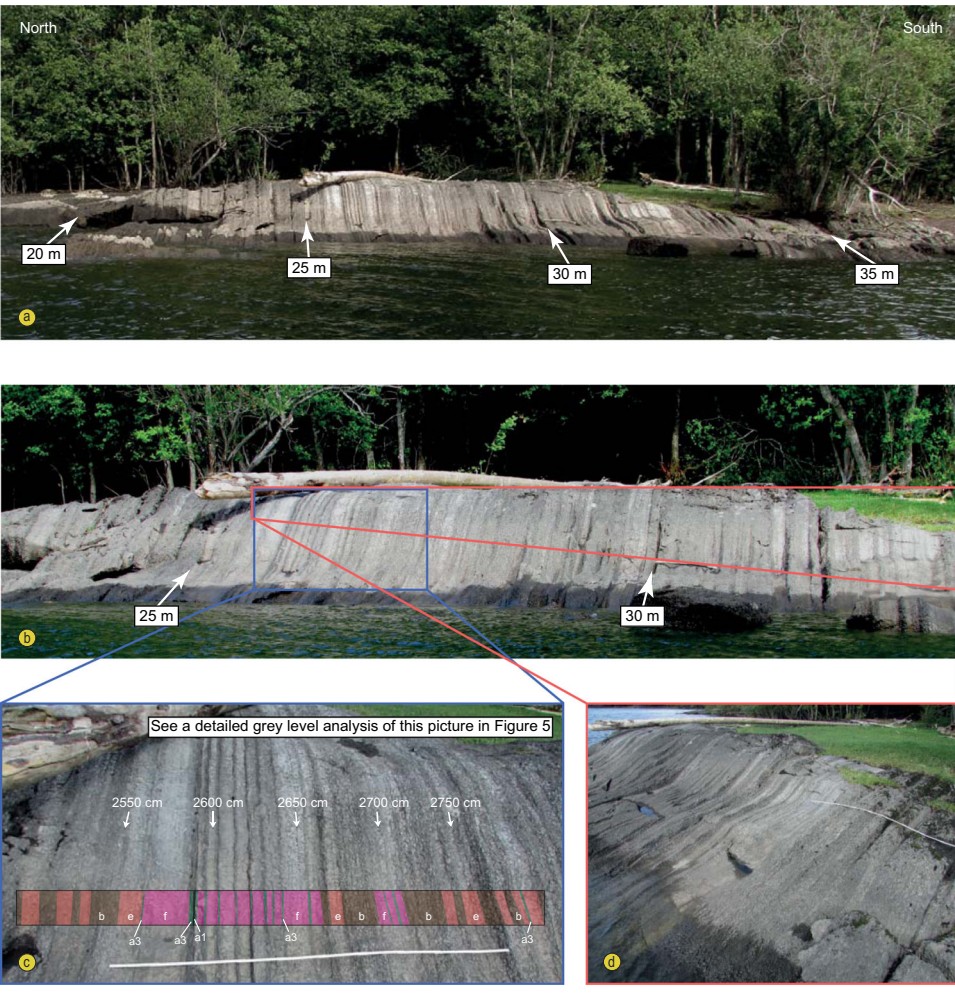

**Fig. 2 Field pictures showing the well-preserved cycles and cycle bundles of the Steinsodden section.** The white ruler in the two pictures at the base is 2 m long. Note the vertical bedding.

The observed sedimentary cycles from the allochthonous Steinsodden section correspond roughly to the cyclic pattern seen in corresponding Middle Ordovician autochthonous limestone sections from a slightly more proximal position on the outer part of the palaeoshelf deposited in Jämtland, Sweden[22], but are clearly different from the nodular and more clay-rich diagenetic rhythmites that are common in the shallower-water Ordovician succession of the Oslo Region farther south[20,23]. Differential diagenesis with redistribution of calcium carbonate, dissolved in soft lithologies and interlayers, and precipitated as cement in limestone beds, can also significantly distort the climatic signal in primarily-deposited marl–limestone alternations[24]. High frequencies can be particularly sensitive to such processes with increasing differential compaction between limestones and interlayers[24]. Metronomic FM analysis (FM-analysis) is well-suited to overcome such distortions because it focuses on the expression of thickness changes over a group or bundle of couplets. The focus on bundles and on the couplet-bundle hierarchy is essential because contrary to primary alternations, changes at the bundle scale have always been related to environmental change[24]. As shown below, the non-random patterns of our FM analysis of cycle thickness point to an unambiguous characterization of Milankovitch cycles at Steinsodden and allow for an astronomical calibration of the section.

**Cyclostratigraphic results**. This lithological rank time-series (see also 'Methods') points to a high significance of four main periodicities at ca. 14, 16 and 19 cm and 283 cm intervals in the Steinsodden section (Fig. 4). The ratio between the two most prominent periodicities at 16 and 283 cm is ca. 1/18, i.e. close to an expected ratio of 1/20 for the 405 kyr eccentricity to precession, assuming that the average duration of the climatic precession in the Ordovician was close to 20 kyr[25]. Moreover, the other significant periodicities at 19 and 14 cm give ratios of 1/15 and 1/20 to the 283 cm main periodicity. Therefore, two of the main interpreted precession periodicities fall close to the expected ratio between precession and 405 kyr cycle of the Ordovician (Fig. 4a). Considering this hypothesis, there is no significant peak in the expected frequency band of the 100 kyr short eccentricity for our lithological rank time-series (Fig. 4b, c, and g).

This is further corroborated by evolutive spectrograms that show that the two significant families of frequencies (14–19 cm and 283 cm) do not strongly overlap through the time-series; the suggested precession shows high power from 0 to 1850 cm and 2900 cm to the top while the suggested 405 kyr frequency shows high power only between 1750 cm and 2750 cm (Fig. 5). Therefore, neither precession nor short-eccentricity appear to be expressed in the latter interval from 1750 to 2750 cm. However, an analysis of a high-resolution grey level signal extracted from the picture of the interval at 2500 to 2800 cm where several cycles appear well-expressed, reveals the expression of both precession and short-eccentricity cycles at 7 to 17 cm and centred around 55 cm, respectively (Fig. 5). This complimentary analysis thus suggests a wider frequency band for precession, in line with

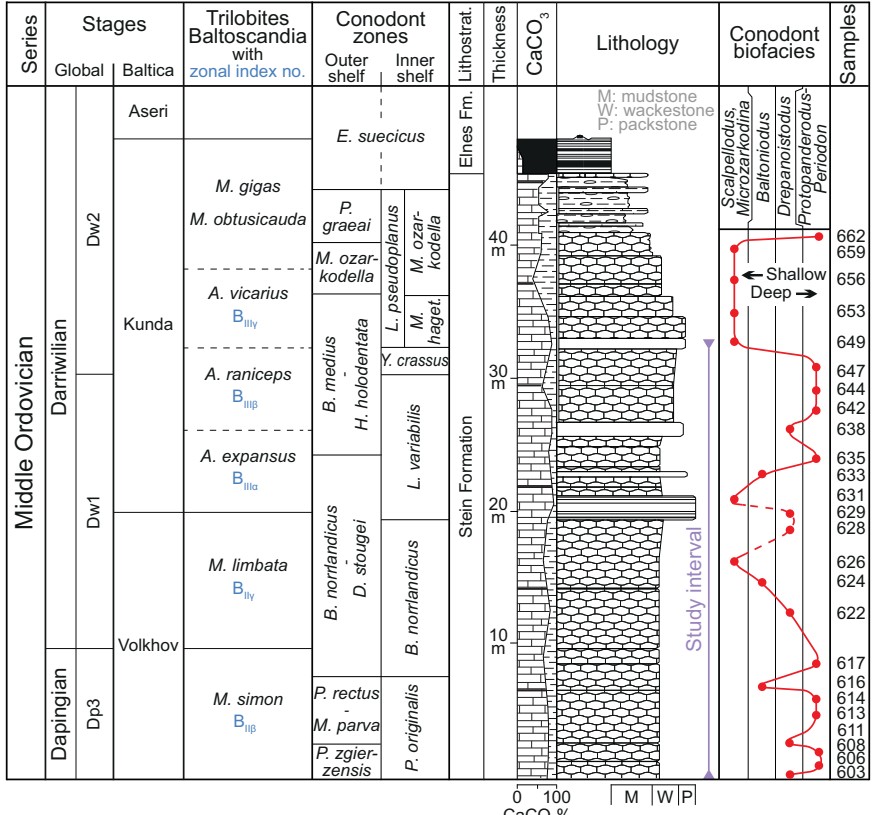

**Fig. 3 Synthetic log, stratigraphy and relative sea-level changes[12,50] estimated for the studied section at Steinsodden.** The trilobite zonation is obtained by correlation to conodont zones established for the section in this study.

**Table 1 Description of the distinct lithofacies recognized in the field with their respective rank used for our time-series analysis.**

| Lithofacies | Rank | Description |
|---|---|---|
| a1 | −1 | Cracks and eroded claystone |
| a2 | −1 | Shale and siliciclastic claystone |
| a3 | −1 | Dark, thin, siliciclastic siltstone seam |
| b | −0.5 | Marly limestone, finely reticulate |
| c | 0 | Marly limestone, coarsely reticulate |
| d | 0.5 | Relatively pure limestone, thin-bedded |
| e | 1 | Relatively pure limestone with few thin clay seams |
| f | 1.5 | Pure massive limestone |

Note that all subfacies of a (a1–a3) are coded with the same rank. Illustrations of various lithofacies are given in the Supplementary Figs. 4–8.

strong frequency modulations and a shift toward slightly lower sedimentation rates in this interval as compared to what is deciphered by our lithology rank time-series. This analysis would point to a ratio of 1–5 (rather than 1–4) between the 405 kyr eccentricity and the short-eccentricity depicted from grey-level variations, which is perhaps due to significant sedimentation rate variations across the identified 100 kyr cycles and the possible expression of one obliquity cycle at 2728–2755 cm that comprises two potential precession cycles as observed from the filter output of this component (Fig. 5). However, visual interpretation from the filter outputs seems to generally match our interpretation of a Darriwilian 'Dar$_{405}$5' long-eccentricity cycle depicted from a 405 kyr filter output of the lithological rank data (Figs. 5 and 6).

In the lower part of the time-series (0–1900 cm), the suggested precession frequency shows periodic bifurcations in power which are typical of the expression of amplitude modulations by the short eccentricity (Fig. 6). To test this further, the FM analysis was conducted (FM, see 'Methods') that further supports our interpretation (Fig. 7). The non-detrended cycle thickness time-series shows two significant peaks of cycle bundling at 1:5.4 and 1:4.6, respectively corresponding to periodicities of 108 and 92 kyr in frequency modulation of an average 20 kyr cycle (Fig. 6b).

In addition, a non-significant 1:16.6 bundling ratio likely reflects the expression of a weak frequency modulation by the 405 kyr eccentricity component. Although this bundling frequency is not significant in the multitaper method (MTM) periodogram of the cycle thickness (Figs. 4f; 7b), a number of 405 kyr bundles can be depicted directly by eye in the time-series (Fig. 7c), as well as in an evolutionary spectrogram of the thickness time series (Fig. 7e). Moreover, metronomic FM spectra of climatic time-series are predicted to show harmonics (positive integer multiples) of the main bundling frequency[26]. A harmonic of the short-eccentricity is indeed present in our MTM periodogram (Fig. 7b). These non-random patterns cannot be produced by diagenesis alone, as has previously been suggested for similar lithological changes[20], but instead point to the unambiguous expression of an orbital control on sedimentation. Subsequently, we applied several tuning approaches including (1) a filter output of the precession frequency (at 0.04–0.1 cycles/cm) from the lithological rank time-series points to between 204 and 212 cycles accounting for a maximum total duration of 4230 kyr for the studied interval if we choose an average duration of 20 kyr for the precession. This tuning approach is compromised by the speculative and over-interpreted cycles from the filter output in

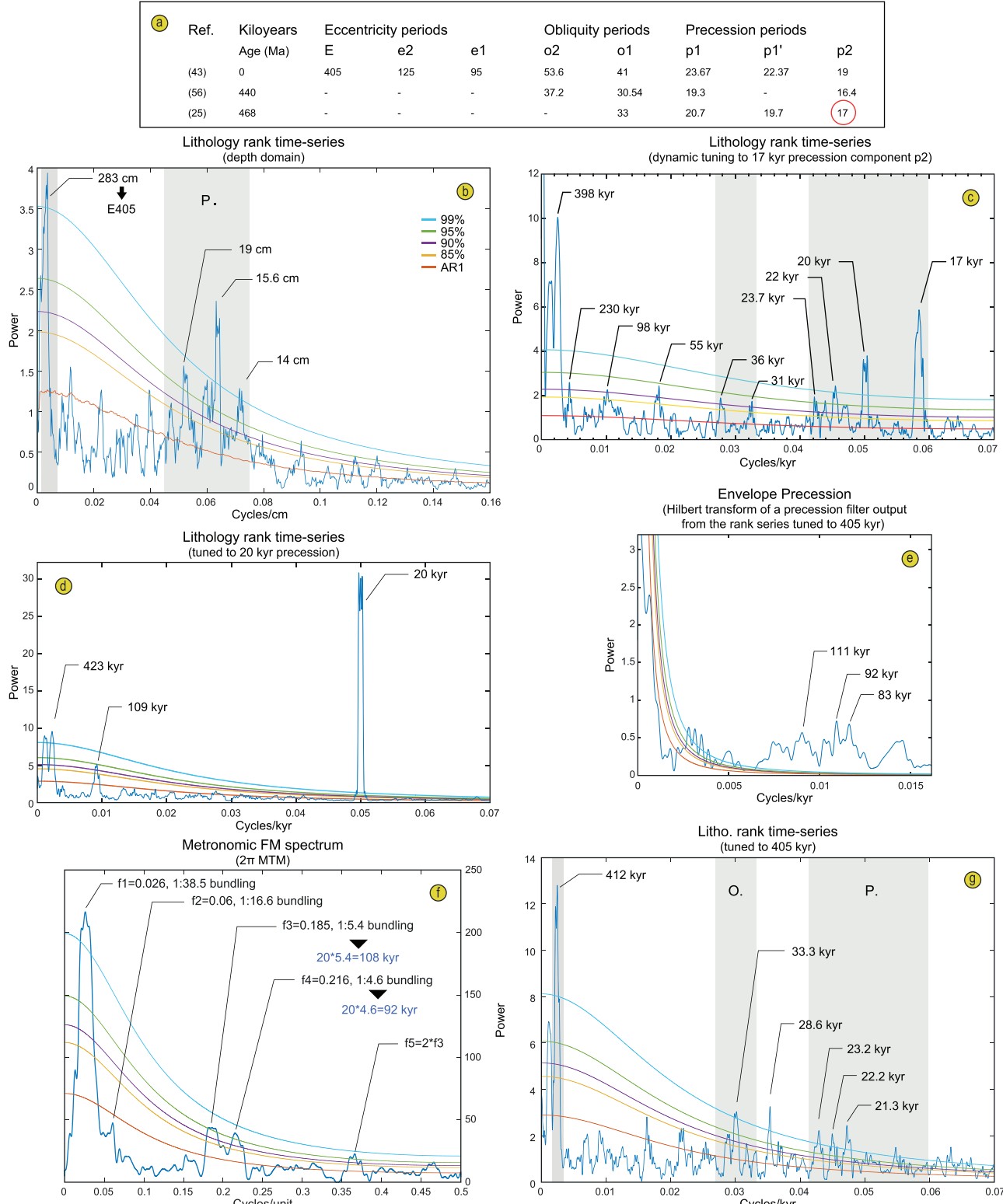

**Fig. 4 Spectral analysis by various 2 π multi-taper method (MTM) periodograms. a** Comparison of main Milankovitch periodicities for the LaO4 astronomical solution for the Recent (see 'Methods') and at 440 and 468 Ma for the obliquity and precession. **b** Periodogram for the non-detrended lithological rank time-series (depth domain). **c** Periodogram of the time-series tuned by frequency stabilization of the 17 kyr precession component (see Supplementary Fig. 10 for details on the frequency stabilization procedure). **d** Periodogram of the rank time-series tuned to precession cycles. **e** Periodogram of the Hilbert transform (Envelope) of the precession filter output extracted from the rank time-series tuned to 405 kyr cycles. **f** Periodogram of the metronomic FM time-series showing bundling at ratios of 1:5.4 and 1:4.6. **g** Periodogram of the rank time-series tuned to 405 kyr cycles.

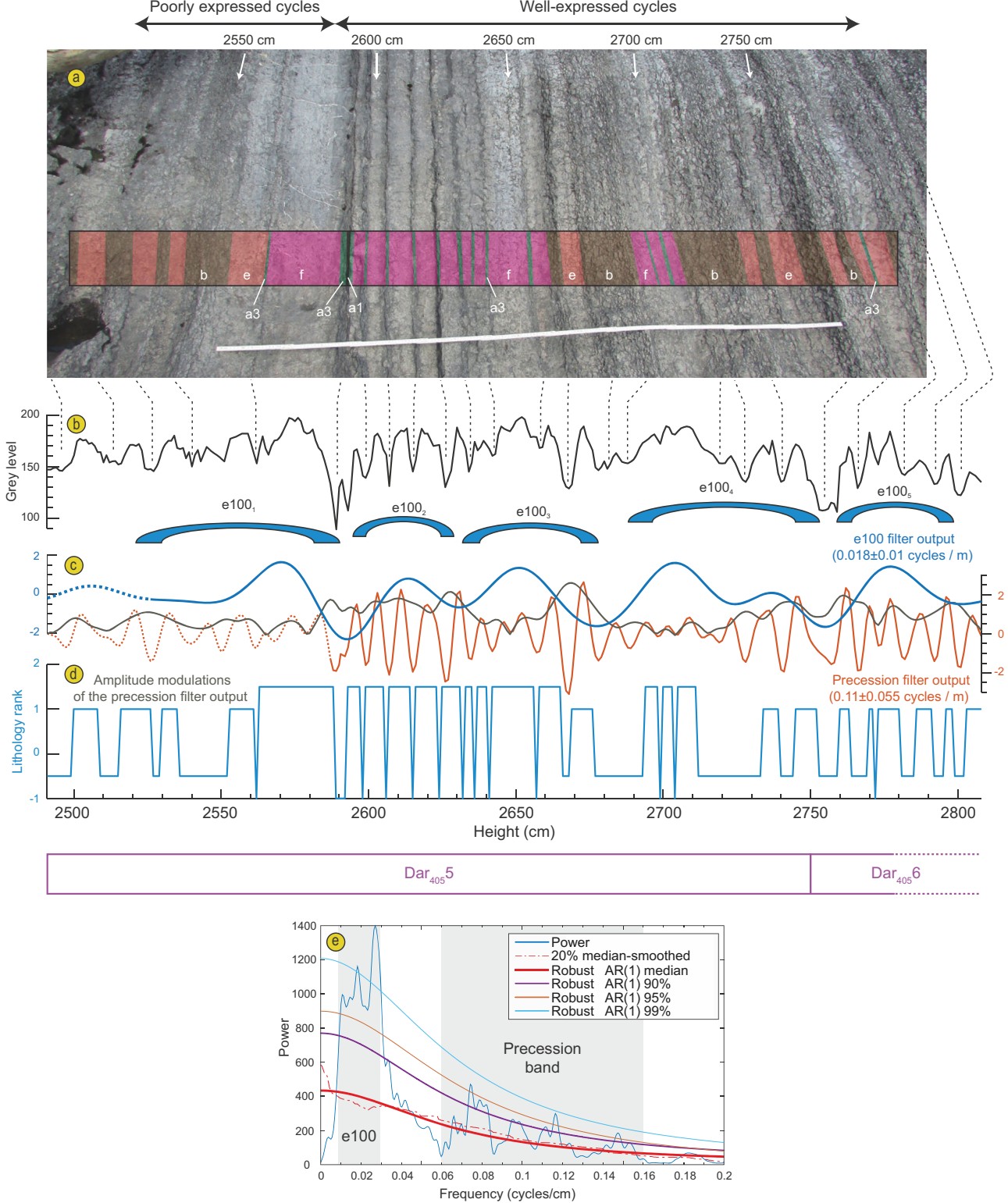

**Fig. 5 Cyclostratigraphic analysis of a grey level signal. a** Photograph showing lithological variations, with corresponding lithofacies of Table 1. **b** Grey level signal. **c** Relevant filter outputs. **d** Lithology rank between 2500 and 2800 cm. **e** 2 pi MTM power spectrum of the grey level signal.

the interval between 2100 and 2600 cm where precession seems very poorly expressed (Fig. 6) but highlights the expression of 100 kyr and 35 kyr components not detected before (Supplementary Fig. 9). (2) A filter output of the 405 kyr periodicity (0.0015–0.0045 cycles/cm) identifies at least eight cycles but does not allow for a full duration assessment due to a very weak

expression of this component between 590 and 1840 cm (Fig. 6). (3) The cycle thickness FM time-series comprises 195 complete primary alternations recognized in the field and the 100 kyr filter output of this time-series points to 38.5 cycles, accounting for durations of 3900 and 3850 kyr, respectively for the section (Fig. 7c). (4) A dynamic tuning of the time-series is based on the

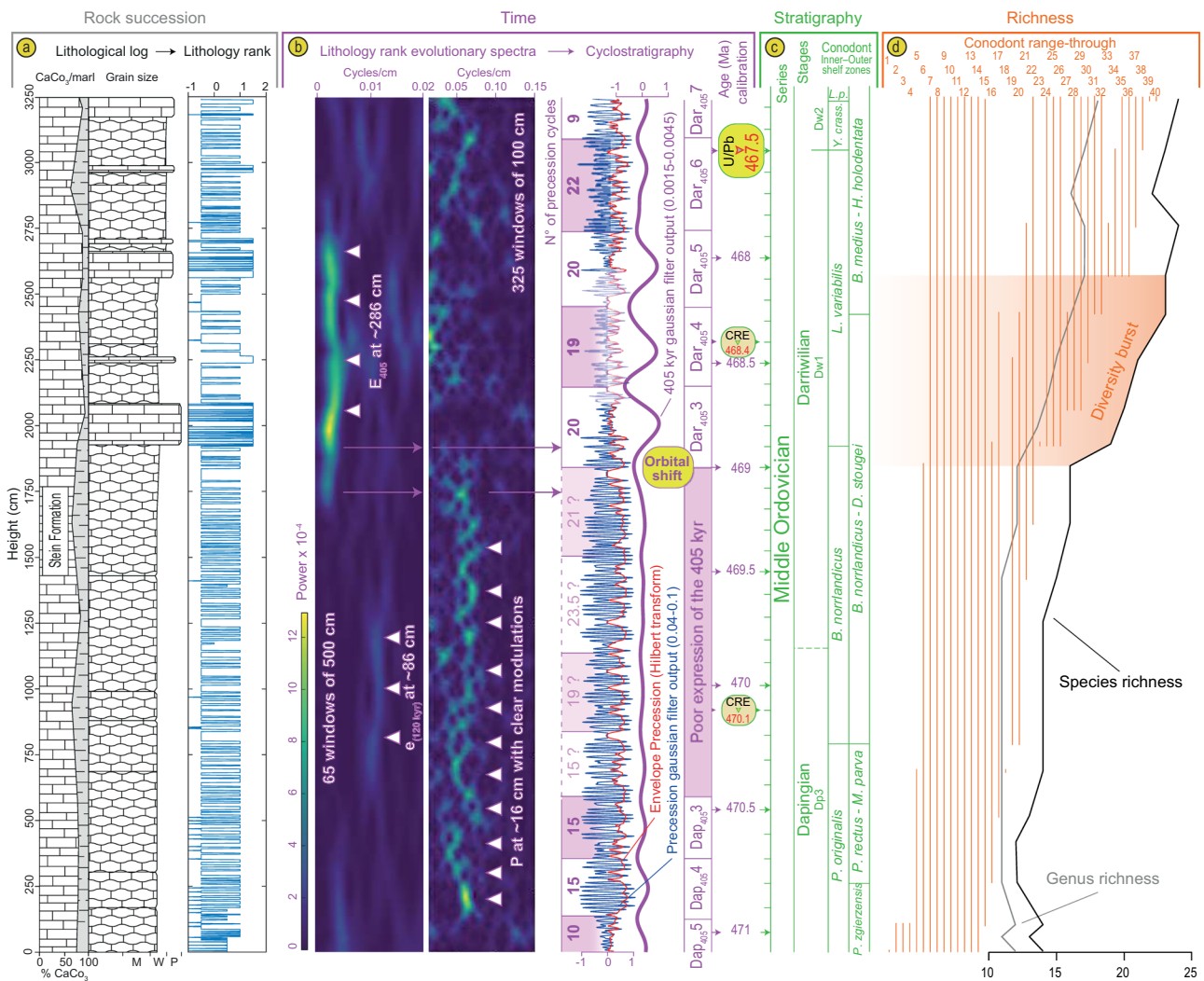

**Fig. 6 Cyclostratigraphic results and conodont ranges and richness. a** Grey column show lithology rank coded against the rock succession (Grey). **b** Pink column show a Time-Frequency Weighted Fast Fourier Transform (TFWFFT) evolutive spectra (see 'Methods') of the lithology rank time-series highlighting the shift from precession- to 405 kyr eccentricity-dominated carbonate sedimentation between 1800 and 1900 cm. The extracted 405 kyr and 20 kyr filter outputs from the rank time-series are based on Gaussian filters (see 'Methods'). Pink and white boxes highlight the distinct interpreted 405 kyr cycles and the respective number of counted precession cycles inside these boxes. Note that the transparent boxes correspond to 405 kyr cycles that are much less pronounced in the 405 kyr filter output and hence less reliable while transparency over the precession filter output highlights a likely overly interpreted precession signal due to the poor expression of this component in this interval. **c** The stratigraphy of the section is shown in the green column. **d** Orange column show the range-interpolated conodont richness (see 'Methods', Supplementary Data 2).

recognition of a powerful 17 kyr component of the precession. Dynamic tuning is an approach where one particular frequency (rather than a wide frequency band), followed along an evolutive FFT spectrum (Supplementary Fig. 10) is assigned to one particular orbital component with its duration determined from the middle age of the studied time-series according to estimated past values of the main astronomical periods proposed in the literature. Our chosen astronomical target component is the P2 peak of the precession cycle which appears particularly well-expressed in our data and has a duration of 17 kyr according to ref. [25] (Supplementary Figs. 10–11). This dynamic tuning approach provides a duration of 3820 kyr (Fig. 6 and Supplementary Fig. 11) and a periodogram with many frequencies that match those of expected orbital components for the Middle Ordovician (compare Fig. 4a–c).

Lastly, we conducted a COCO procedure that considers the correlation between the detected frequency peaks (see 'Methods' section and Supplementary Fig. 12). This approach points to a range of average sedimentation rates of 0.7–0.85 cm/kyr for our

cycles to reflect Milankovitch components, with this range allowing us to calculate cumulative age uncertainties and with the sedimentation rate at 0.85 cm/kyr fitting our favoured dynamic tuning approach. This latter approach corresponds to the minimum duration estimate of 3820 kyr for our complete section. Considering the latter tuning approach and taking the base *Y. crassus* as our radiometric anchor, we estimate a maximum cyclostratigraphic uncertainty of +0.24 Ma for the base of *L. variabilis*, +0.56 Ma for the base of *B. norrlandicus*, +0.70 for the base of *M. parva* and +0.76 Ma for the base of *P. originalis* in the section (cumulative uncertainties are calculated from the top as the difference between a minimum duration estimate at 0.85 cm/kyr and maximum estimate at 0.7 cm/kyr).

**Astronomical tuning of the GOBE.** We used conodont zonation of the section to anchor our astrochronologic time scale to a U-Pb date of 467.5 ± 0.28 Ma at the base of the *Yangtzeplacognathus crassus* Zone in the Kinnekulle area, southern Sweden[10]. The

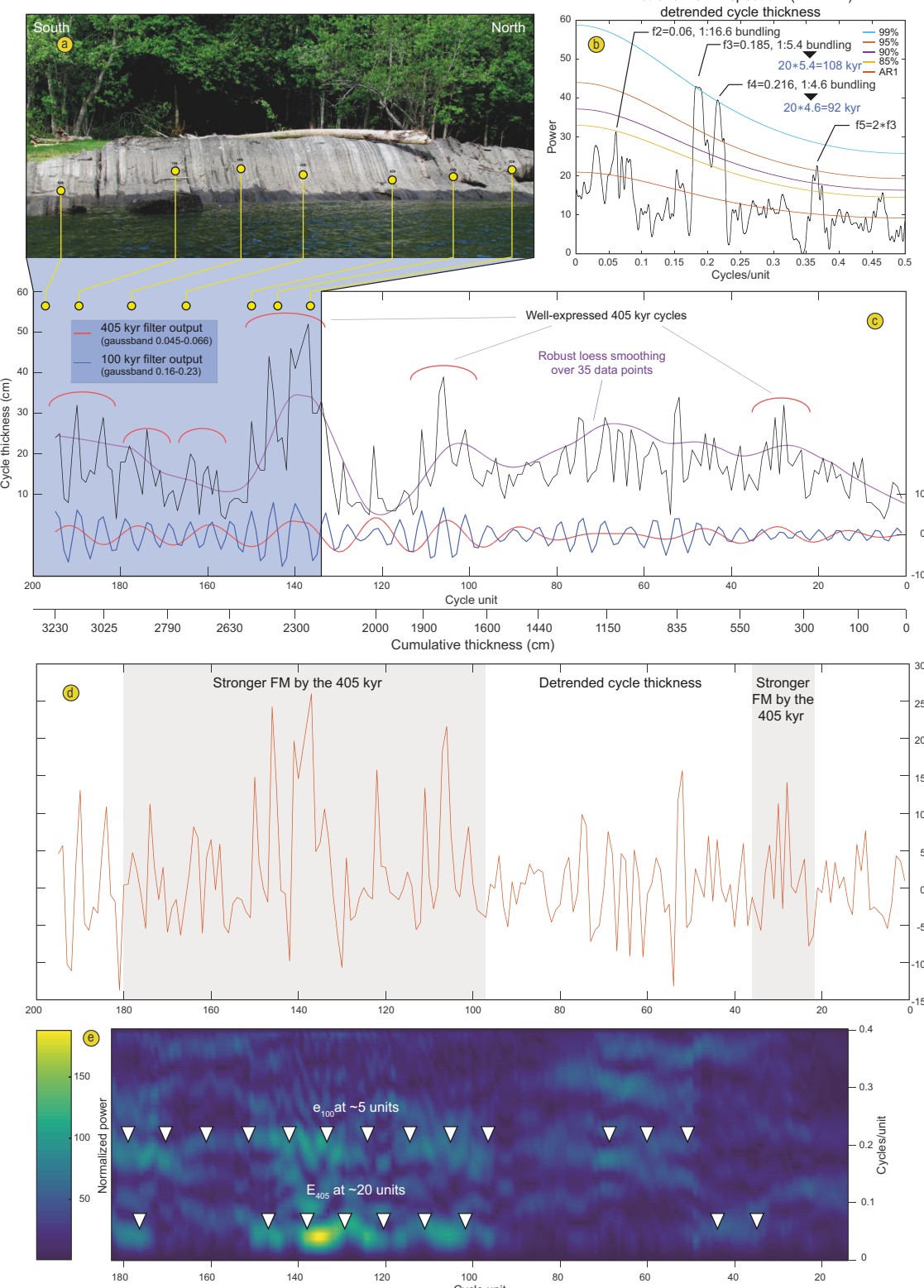

**Fig. 7 Metronomic frequency modulation analysis. a** Close-up photograph of the 2250–3250 cm interval (mirrored). **b** 2 π MTM periodogram of the detrended FM time-series showing highly significant bundling frequencies corresponding to 92 and 108 kyr cycles when primary alternations are set at 20 kyr. **c** Cycle thickness FM time-series along with the 100 kyr and 405 kyr filter outputs. **d** detrended FM time-series. **e** Evo-FFT of the detrended time-series showing significant frequency modulation around 5 units (short eccentricity band) as well as a significant frequency modulation at ca. 20 units (405 kyr) in restricted parts of the time-series, between 1600 and 3000 cm. Another short interval at the base between 270 and 510 cm also shows a stronger FM by the 405 kyr cycle.

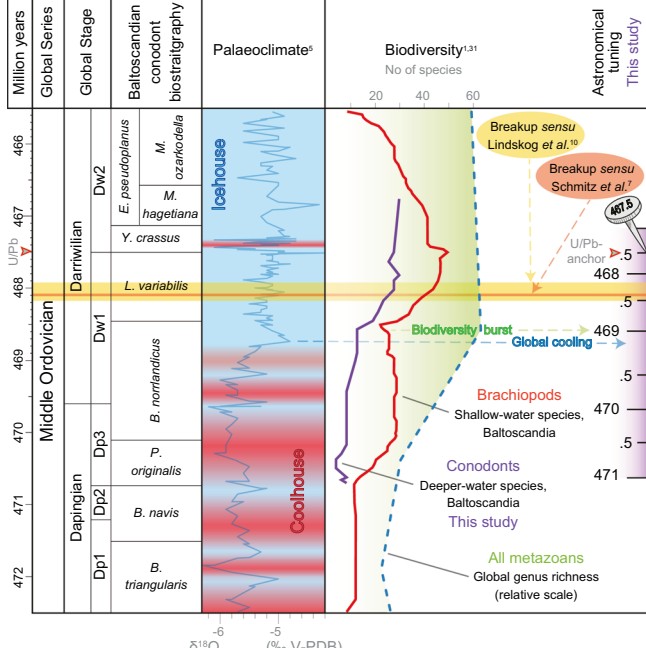

**Fig. 8 Middle Ordovician geochronology[10], conodont biostratigraphy[13], palaeoclimate[5] and biodiversity[1,31].** The astronomical tuning and conodont richness added by the current study is marked in purple. The astronomical timescale is anchored in the high-precision zircon U/Pb date of 467.5 ± 0.28 Ma[10]. Sketched palaeoclimate trends highlight several punctual cooling episodes in the Dapingian coolhouse before the major Darriwilian cooling. The global onset of the GOBE coincides with regional richness spikes across clades and facies belts. Note that the asteroid breakup is well-constrained[7,10] and postdates both palaeoclimatic and biotic events.

Kinnekulle section is characterized by condensed cool-water carbonate facies that was deposited in the shallow-water epi-continental sea known as the Baltoscandian Palaeobasin[27]. This basin was characterized by slow carbonate production and an extremely low relief[28] that resulted in very little siliciclastic input. During the upper half of the lower Darriwilian *L. variabilis* Conodont Zone unusual concentrations of micro-meteorites from the L-chondrite breakup event has been reported[8] and from the lowermost bed of the overlying *Y. crassus* Conodont Zone euhedral zircons demonstrated to have a volcanic origin has been described yielding the above-mentioned U-Pb age[10]. Cosmic ray exposure (CRE) ages were calculated based on the stratigraphical occurrences of the micro meteorites and the CRE-ages were subsequently anchored in the high-precision U-Pb age thereby resolving the Kinnekulle succession in absolute time[10]. This mid-Ordovician time-scale derived from CRE-ages was recently challenged by recent U-Pb dating of zircon populations obtained from the same limestone bed as refs. [10,29]. A Bayesian age of 465.01 ± 0.26 Ma was proposed for this stratigraphic level, derived from the youngest population of overlapping zircon grains. Contrary to the age obtained in refs. [10], the recent Bayesian age for the base of *Y. crassus* is at odds with the most recent Ordovician time-scale[30]. Propagation of the age of ref. [29] downward provides an age of 469.6 ± 0.3 Ma for the Floian–Dapingian boundary. This contradicts the spline-fit ages of the base Dapingian at 471.3 Ma for a Graptolite composite and 473.9 ± 1.21 Ma for the Conodont composite of the most recent Geologic Time Scale[30]. For this reason, we have favoured the older CRE age-scale and its anchor of base *Y. crassus* at 467.5 ± 0.28 Ma but in the absence of any certainty in the

numerical ages of the Middle Ordovician time-scale, all ages obtained here can easily be shifted downward by 2.5 Ma if the revised age of ref. [29] is proved correct in the future. It is note-worthy that besides their disagreement on absolute ages, the two studies seem to agree on the average sedimentation rate to be applied to the dated section down into the Darriwilian and Dapingian and hence the duration of conodont zones should remain the same[10,29].

The age-scale derived from the Kinnekulle record thus provides an independent control to which we can confront our results from the Steinsodden cyclostratigraphy. In addition to the U-Pb date cited above for the base of *Y. crassus*, the Kinnekulle age-scale provides an average age of 468.42 Ma for the base of *L. variabilis* and an average of 470.1 Ma for the base of *B. norrlandicus* using ref. [10]'s favoured 4 mm/kyr sedimentation rate. This much lower sedimentation rate at Kinnekulle compared to the one we derived for Steinsodden could be related to more distal conditions in the latter section that favoured continuous sedimentation.

A relative time scale for the Steinsodden section obtained from the dynamic tuning based on the identification and picking of the 17 kyr component of precession has been anchored to the U-Pb date of 467.5 ± 0.28 Ma allowing an overall match of the U-Pb-derived timescale to our astronomical clock (Fig. 6). Taken all together, our data point at an average duration of 4025 ± 205 kyr (3820 kyr in our favoured astronomical dynamic tuning) covering the interval from the base of *P. originalis* to the very base of the *L. pseudoplanus* conodont zones whereas the average scale based on the Kinnekulle record accounts for 3620 kyr[10] for the same interval (Fig. 8). This estimate contrasts with the maximum duration of 4640 kyr derived from a minimum sedimentation rate of 0.7 cm/kyr given by the COCO procedure and is thus in favour of our interpretation of a much shorter interval around 3820 kyr with an average sedimentation rate of 0.85 cm/kyr (maximum compatible sedimentation rate of the COCO procedure). Our tuning points at a duration of ca. 228 kyr for the *Y. crassus* Zone (11.5 precession cycles, Fig. 6) at Steinsodden, while estimates from the Kinnekulle record point at a duration in the range of 167–439 kyr based on cosmic-ray exposure ages[10]. Moreover, ages derived from the dynamic tuning for the base of *L. variabilis* (468.9 Ma) and base *B. norrlandicus* (470.25 Ma) fall within an error of 500 kyr and 150 kyr, respectively, from the Kinnekulle CRE-derived ages[10].

A conodont range-interpolated richness estimate from the section (Figs. 6 and 8) delineates a rapid increase in species across the transition from the *B. norrlandicus* Conodont Zone to the *L. variabilis* Conodont Zone, followed by a sustained radiation reaching up to 17 genera and 23 species at 2577 cm in the middle part of the *L. variabilis* Conodont Zone (Fig. 6). The bulk of this radiation occurs within just a few hundred thousand years, starting at 469 million years ago according to our astronomical time scale (Fig. 6). This mirrors the trend seen in shallow-water brachiopods faunas from the East Baltic[31], calibrated to the same age-scale (Fig. 8). The radiations thus occurred coincidently within nektonic and sessile benthic organisms across facies belts with the onset of a steep rise from the base of the *L. variabilis* Conodont Zone. This rise in richness coincides with the prominent shift from precession to a 405 kyr eccentricity-dominated cyclicity (Figs. 6 and 8).

## Discussion

Since the GOBE was first defined as an event spanning the entire Ordovician[32], a considerable body of literature has now con-strained the majority of the biotic radiations to start within the early Darriwilian[33]. This revised view compiled at various

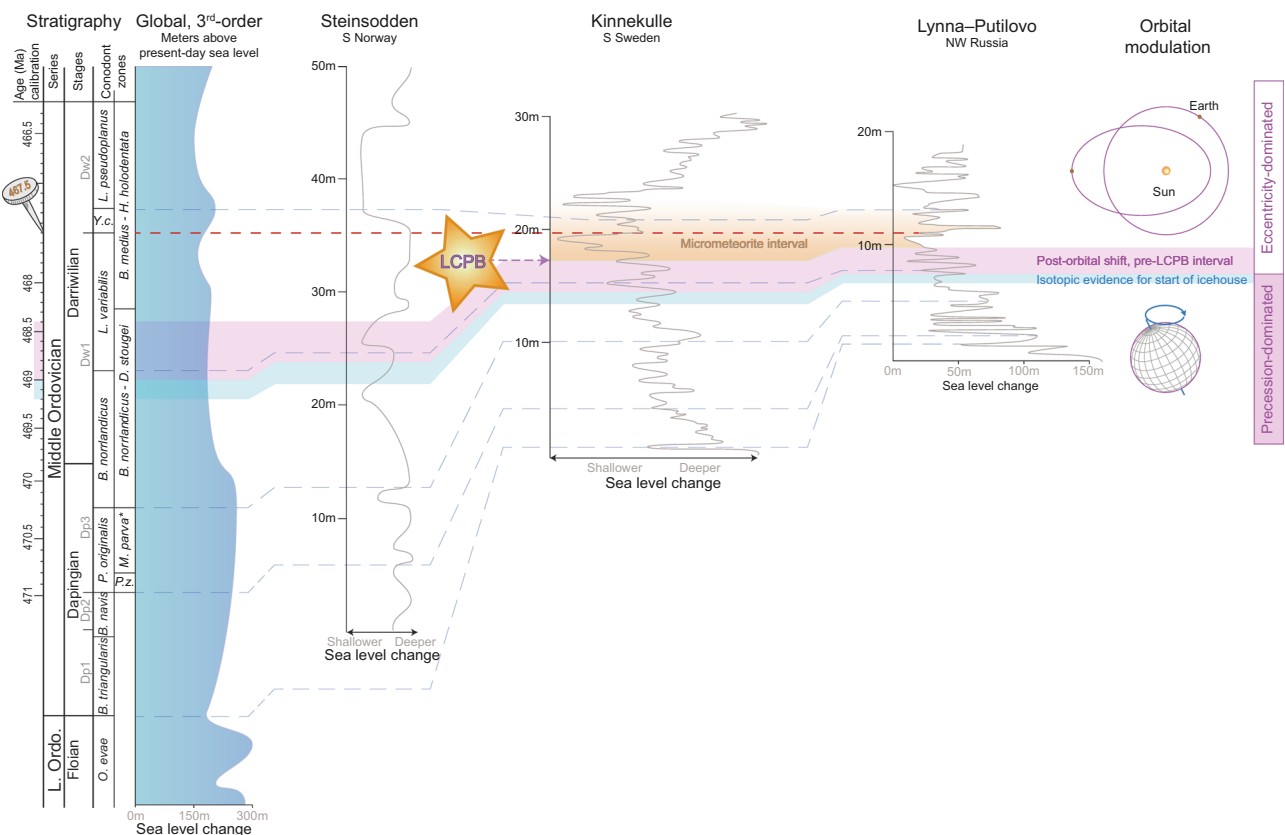

**Fig. 9 Regional sea level evolution across the eastwards thinning Middle Ordovician Baltoscandian Palaeobasin[5,27,50] and the global trend[1,35].** Each section is anchored in the U-Pb-age known from the lowermost bed of the *Y. crassus* Conodont Zone[10] (red-dotted line). Sea level fluctuations are correlated at the biozonal level and the temporal position of the L-chondrite parent body break up (LCPB) and the micrometeorite-bearing intervals in Sweden and Russia are shown to scale. The shift from precession- to eccentricity-dominated orbital modulation is shaded in pink and the preceding stratigraphical interval where isotopic evidence shows a shift to icehouse conditions is shaded in blue.

taxonomic levels appears nearly synchronous globally and is also in line with Jack Sepkoski's classic global metazoan compilations[6,34], as well as recent temporally better resolved estimates[1,3].

The GOBE has been speculated to be fuelled by a shift from a greenhouse state in the Early Ordovician Floian Age to an icehouse state in the Darriwilian[4,5]. Oxygen isotope analyses of well-preserved brachiopods from the Ordovician of Baltoscandia[5] show a progressive decrease in temperature and increase in seawater $\delta^{18}O$ through the Dapingian, punctuated by several pronounced short-lived cooling pulses (Fig. 8). Both regional and global sea level curves (Fig. 9) support this notion of a Dapingian "coolhouse" interval as a large sea level drop is found at the Early–Middle Ordovician boundary[35,36]. This sea level drop, which is estimated to be in the order of 70–80 m[5], is reflected in the rock record as a well-developed hardground surface seen throughout Baltoscandia[27,37]. In our view, this surface reflects that the earliest Middle Ordovician Dapingian Age was a forerunner to the cooling trends that followed with the establishment of much colder climate in the Darriwilian, coinciding with the main phase of the GOBE.

The claim that dust from the LCPB-disruption caused the mid-Ordovician ice age[7] was proposed following the observation that the first occurrence of extra-terrestrial chromites coincides with the onset of a major sea level lowstand within the *L. variabilis* Conodont Zone in the Kinnekulle area. This lowstand is also known as the Täljsten level. However, this sea level drop that should supposedly reflect the start of the ice age is only one among several within the *L. variabilis* Conodont Zone, across

numerous sections of the Baltoscandian Palaeobasin[5,27] (Fig. 9). In contrast, the orbital shift from precession to 405 kyr eccentricity cycles observed here is located in the topmost part of the preceding *B. norrlandicus* Conodont Zone and its onset also coincides with a significant sea-level fall that occurred earlier than the one depicted in coincidence with the onset of the micrometeorite interval (Fig. 9).

Our study thus deciphers the exact timing and sequence of events through the early Darriwilian as follows (Fig. 8): Significant cooling first occurred around 469.2 Ma in the top of the *B. norrlandicus* Conodont Zone as marked by a prominent positive shift in brachiopod calcite $\delta^{18}O$. Then a major rise in brachiopod and conodont richness followed ~200 kyr later at 469 Ma in coincidence with our orbital shift from precession dominance to 405 kyr eccentricity. The base of the micrometeorite interval that marks the timing of the LCPB projects, on our timescale, at 468.4 Ma, postdating the onset of glaciation by 800 kyr and the onset of the GOBE by 600 kyr. Hence, the LCPB cannot have been a facilitating factor, neither in modulating climate, nor in the GOBE. In fact, the timing of the LCPB immediately precedes an interval where both the conodont and brachiopod species richness curves become less steep, suggesting a decrease in speciation rates as compared to pre-LCPB conditions (Fig. 8). Therefore, rather than sparking the GOBE, the dust rapidly delivered from the asteroid breakup more likely acted as a temporary brake on biodiversity accumulation.

Glacially associated precession–obliquity and obliquity–eccentricity orbital switches are known from Cenozoic subtropical marine carbonate records[38,39]. However, our Palaeozoic analogue, characterized

by a change in orbital beat from precession to 405 kyr eccentricity, has not been depicted before but likely results from energy transfer from high (precession) to low frequencies (long-eccentricity) via a direct response to eccentricity modulations. Such energy transfers have been illustrated for the Oligocene–Miocene interval and likely result from significant changes in ice volume[40]. This orbital expression may also be related to a change in ocean chemistry, or changing response of the carbon cycle, as various carbon reservoirs (such as the ocean dissolved inorganic carbon and marine organic carbon) are particularly sensitive to orbital climate change and exert a strong influence on global climate. However, the Middle Ordovician orbital energy transfer occurs during a quiescent interval with respect to carbon isotopes, preceding by 1.7 million years a prolonged positive carbon isotope excursion (MDICE)[5]. Neither does the Steinsodden section reflect a change in sedimentation rate. Thus, an intrinsic climatic response must be invoked to explain this event in carbonate sedimentation, involving a particular orbital configuration or an intrinsic Earth reservoir that significantly dampens the power of precession and subsequently enhances that of the long-eccentricity.

Nodes of 2.4 million year Milankovitch grand cycles that modulate eccentricity can significantly dampen precession power in favour of other orbital components[39,41], while at the same time being responsible for global cooling and ice sheet expansion due to minimum variability in insolation changes[39,42]. In the Cenozoic, the Eocene/Oligocene transition (EOT) shows a well-established shift from a greenhouse to icehouse mode that occurred at the time of a near conjunction of nodes in 2.4 myr and 1.2 myr grand cycles[43,44]. During the Neogene, long-period mammalian turnover pulses coincide with 2.4 myr eccentricity and 1.2 myr obliquity nodes showing a significant influence of these peculiar orbital configurations that favour global cooling and hence trigger associated biodiversity changes[45]. An early Palaeozoic example shows that over a time span of 60 myr, the variance in biological turnover of graptoloids could be explained by insolation changes associated with 2.4 myr and 1.2 myr grand cycles[46].

Changes in ice volume and the timing of a node in 1.2/2.4 myr grand cycles could be the potential triggers of our orbital energy transfer in carbonate sedimentation of the Steinsodden section. The amplitude of change in benthic $\delta^{18}O$ at low latitudes across the EOT is in the order of 1‰ which is of a similar amplitude to what is observed from the Dapingian to the Darriwilian[5,44]. The growth in volume of the East Antarctic ice sheet (EAIS) strongly influenced the expression of orbital cycles recorded in southern high-latitude clastic sediments[42]. Prior to the Oligocene glacial maximum, the growth and volume of a small-sized EAIS was primarily paced by the influence of obliquity and precession whereas after the establishment of a large-sized EAIS, ice volume was paced by eccentricity with a much enhanced 405 kyr cycle[42]. The orbital beat of such ice-volume changes influences global climate through associated changes in the carbon cycle, in sea-level and in the calcite compensation depth that control calcium carbonate accumulation rate and, hence, the percentage of carbonates in marine sediments as recorded in our lithological rank time-series. In the evolutive power spectrum of our lithological rank time-series tuned to the precession, we observe the sudden expression of the obliquity component shortly after the energy transfer from precession to 405 kyr, which is in strong contrast to the rest of the section where it is absent (Supplementary Fig. 9). This observation is once again similar to observations made across the Eocene–Oligocene transition interval[42,44,47] and generally, reflects the conjunction of a 405 kyr minimum and a node of the s4–s3, 1.2 myr amplitude modulation of obliquity, a configuration that favours cooling and ice sheet expansion in the Cenozoic and influences biodiversity[45,48]. In such a peculiar orbital configuration, the amplitude of the eccentricity-modulated precession is dampened because of a very low amplitude and power in eccentricity, which in compensation, favours a greater expression of the power and amplitude of the obliquity that is generally low and/or poorly expressed at low latitudes.

The orbital climate shift that we record here in our dataset is likely not linked to the shift from a predominant s4–s3 ~1.2 myr grand cycle to a g4–g3 2.4 myr grand cycle demonstrated as a great influence in graptoloid diversity and occurring later than the timing of our investigated interval, around the Darriwilian to Sandbian transition[46]. Our orbital shift from precession to 405 kyr eccentricity could possibly correspond to the conjunction of minima in the 1.2 myr obliquity and 2.4 myr eccentricity grand cycles demonstrated to take place in the Tarim Basin (NW China) at ca. 4.07 myr prior to the Darriwilian/Sandbian boundary and claimed responsible for cooling and rise in graptoloid richness in this area[15]. This conjunction of grand cycles minima within or below the North American *Histiodella sinuosa* conodont Zone correlates to a rather long interval in our study comprising the *L. antivariabilis*—base *Y. crassus* Baltica conodont zones[30]. However, Fang et al.[15] suggest that this orbital event would postdate the base Darriwilian by ca. 2.1 myr whereas our findings place our orbital shift within the first one million years after the base of the Darriwilian. Alternatively, our main orbital shift could also correspond to the 1.2 myr obliquity minimum of ref. [15] that postdates the base Darriwilian by ca. 800 kyr and is in near-conjunction with a 2.4 myr maximum. The comparison of our results to the study by Fang et al.[15] remains tenuous, however, due to the large uncertainty in the exact position of the base Darriwilian in both our study and theirs and in the correlation of our conodont zones to those of the Tarim Basin. Nevertheless, at mid-latitudes, such as that of Baltica during the Middle Ordovician (~40°S), even minor obliquity changes could influence climate[41]. The major Middle Ordovician energy transfer from precession to 405 kyr eccentricity could therefore represent the response of the Earth's climate system to a significant increase in ice volume, favoured by a peculiar orbital configuration such as the node of 1.2 and/or 2.4 myr grand cycles. This situation would be similar to that at the Eocene/Oligocene glaciation, both events that led to increased speciation rates.

## Methods

**Biostratigraphical correlation and lithofacies**. The top of the Stein Formation corresponds to the basal part of the *E. suecicus* Conodont Zone and is overlain by 2.7 m of clays and shales from the Elnes Formation, corresponding to the top of the exposure[13] (Fig. 3). The conodont zones are representative of the outer shelf zonation but common species between outer and inner shelf sections of Baltica and several shelf edge sections globally, have allowed for a precise correlation to the standard inner shelf Baltoscandian conodont zonation[13] and to Baltoscandian stages and global stage slices (Fig. 8). We use a U-Pb date of 467.5 ± 0.28 Ma at the base of the *Y. crassus* Zone[10] to anchor our astrochronologic time scale to the conodont zonation.

The first 32.4 m of the whole section are exposed well-enough to perform a high-resolution sedimentological description of the bedding pattern. Logging and characterization of the lithofacies was performed at the cm scale. Thin section studies of selected sample levels were used as additional methods for the lithologic determination[13]. The six main lithofacies were recognized conspicuously in the field (Table 1) and were assigned a digital rank with low values characterizing more siliciclastic-rich lithologies and high values characterizing pure limestones. These ranks were assigned for each cm along the section, thus producing a discrete variable through a time-series of 3242 data points. Petrographic studies reveal that bioturbation took place consistently and evenly throughout the studied section, but no identifiable trace fossils were recorded either in the field or in thin sections[13]. Although lithofacies rank and bed thickness may be considered as subjective proxies that potentially bear lower signal to noise ratio than a continuous signal, they have shown great success to decipher orbital cycles[49] and they present the advantage to be particularly sensitive to frequency modulations[26]. Therefore, while our division in lithofacies of same values perhaps hampers the recognition of the full spectrum of orbital components as compared to a continuous digital signal, it presents the advantage to allow for assessing variations in the thickness of couplets, which, if controlled by precession, should show frequency modulations by the eccentricity.

**Conodont ranges and diversity**. The conodont data derive from a large mono-graphic work by one of the present authors[13] with taxonomic revision applied here. A total of 23 investigated samples recorded the ranges of 40 identified conodont species along with the number of specimens in Supplementary Data 2. We high-lighted in red the assumed range-through record of all species as presented in Fig. 6 (absences of species between first and last appearance are treated as presence). The range-through records of all species was built upon their presence either in the studied interval, or in the nearby Herram section situated immediately below in the stratigraphy[13] (species No 11, 12 and 13, Supplementary Data 2), or from a presence noted below in the stratigraphy in other areas of Norway[13] (species No 2 and 8, see Supplementary Data 2), or above in the stratigraphy in the uppermost 10 m of section at Steinsodden not studied for cyclostratigraphy (species No 9, 12 and 31, Supplementary Data 2). Reported ranges and the richness estimates both derive from the range-through assumption.

**Relative sea-level changes across Baltica**. Variations in relative sea-level deli-neated for the Steinsodden section were derived from combined data on conodont biofacies analysis[12,50], sedimentological analyses including thin section studies, major element analyses (atomic absorption spectrophotometry) and the nature of associated faunas, primarily gastropods, ostracods and echinoderms (Fig. 3). The overall biofacies dataset was composed of c. 12,000 conodont specimens of eight selected genera from 76 samples. The samples were collected from Dapingian and Darriwilian strata of the Stein Formation along the Norwegian and Swedish Caledonian nappe front. Multivariate statistical techniques (primarily Correspondence Analysis) revealed four distinct biofacies: *Scalpellodus-Microzarkodina* Biofacies, *Baltoniodus* Biofacies, *Drepanoistodus* Biofacies and *Protopanderodus-Periodon* Biofacies representing gradually deeper or more distal conditions near the platform margin[50]. Each of the analysed samples from the Steinsodden section were placed in one of the four biofacies, delineating biofacies evolution and subsequently a sea-level curve was drawn between them (Fig. 3).

Correlation across the Baltoscandian Palaeobasin at the conodont subzone level is possible as the regional conodont and trilobite zonations for the Middle Ordovician are very well tied to each other, as well as to that of graptolites and isotope stratigraphy[5,10,51,52]. The regional chronostratigraphy for Baltica was based on trilobites during the Middle Ordovician, with the Dapingian–early Darriwilian interval being divided into the Volkhov and Kunda regional stages. The classical trilobite substages have index numbers where $B_{II}$ denotes Volkhov and $B_{III}$ denotes the Kundan. The substage boundaries $B_{II\beta}/B_{II\gamma}$, $B_{II\gamma}/B_{III\alpha}$, $B_{III\alpha}/B_{III\beta}$, and $B_{III\beta}/B_{III\gamma}$ of Baltica correlate approximately with the *P. originalis/B. norrlandicus, B. norrlandicus/L. variabilis, L. variabilis/Y. crassus* and *Y. crassus/L. pseudoplanus* inner shelf conodont zonal boundaries (Fig. 3). The use of these tie-points allowed us to provide the approximate position of the Dapingian/Darriwilian stage boundary in our section which is close to the top of the lower third part of *B. norrlandicus* (Supplementary Fig. 13). This stratigraphical framework further enables the tracking 3rd–5th order sea-level oscillations across the basin (Fig. 9). From the far West where the marginal, expanded Steinsodden section is located over the central part of the basin some 300 kilometres to the southeast where detailed bed-by-bed microfacies analyses from Kinnekulle, South Sweden, are available[27] to the St. Peterburg area, Russia some 1200 km farther East where bed-by-bed biofacies analyses were conducted on more than 50,000 trilobites and brachiopods[5,53], these sections show an intricate hierarchy of sea level change through the Dapingian–Darriwilian interval. Even though the sections get progressively more condensed towards the East, the high temporal resolution of these studies from the shallow-water platform carbonates allows for the tracking of transgressive and regressive pulses as they occur on a stable intra-cratonic setting. The Russian sea level curve is further scaled based on the basin-wide tracking of limestone tongues protruding into siliciclastic deeper-water facies during times of maximum lowstand. This provides an additional constraint on the size of the individual sea level oscillations. Lastly, the Russian dataset directly tie the biofacies data on the bed-by-bed scale to brachiopod $^{18}O$-stable isotope record enabling a paired palaeoecological and geochemical climate proxy. This constrains well the coincidence of the change in dominance from precession to 405 kyr cycles to the onset of a transient 1‰ positive excursion in brachiopod oxygen isotopes interpreted as the onset of the Middle Ordovician ice age[5]. In context of the LCPB-disruption, correlation of the data from Rasmussen et al.[5] to the recent data of Schmitz et al.[7] shows that the onset of the cooling event (and hence the observed energy transfer from precession to 405 kyr eccentricity) are situated below the Volkhovian–Kundan regional stage boundary whereas the base of the interval that contains the first fossil meteorites are situated above this boundary within the lower Kundan (Supplementary Fig. 14). Therefore, it is unlikely that the meteoritic bombardment is the direct cause of this cooling. However, as this event occurred soon after an orbital configuration that favoured the onset of an ice age, it may possibly have intensified icehouse conditions by the dust feedback postulated by Schmitz et al.[7].

**Cyclostratigraphic analysis**. For time-series analysis, the resulting signal was tested using multitaper method spectral techniques (MTM) with robust red noise modelling and three $2\pi$–tapers[54] and Time-Frequency Weighted Fast Fourier Transform[55] (TFWFFT) which is an evolutive spectrogram that can potentially highlight amplitude modulations and shifts through time in the dominant

frequencies (Fig. 6). Metronomic Frequency Modulation Analysis (FM, e.g. ref. [26]) was specifically designed for testing discrete signals defined by lithological ranks in sections that show well-defined primary alternations and bundling of the primary cycles (Fig. 7). The FM analysis relies on the principle that stratigraphic cycle thickness is proportional to cycle duration. If primary alternations represent the precession, then cycle thickness bundling should be linearly related to the frequency modulation of the precession, i.e. to the orbital eccentricity (ca. 100 kyr and 405 kyr). A number of bundles were evident in the field (Fig. 2) and we noticed variations in thickness of the primary alternations that justified testing a FM analysis on our dataset. We used the procedure of Herbert[14] that consists in the construction of a time-series where each primary alternation or cycle identified in the field is measured for its thickness and assigned to a cycle unit (Supplementary Data 3). This new time-series was then tested with a $2\pi$–MTM periodogram with the frequency expressed in cycles/unit. All frequencies of interest were extracted from the signals using a Gaussian filter with user-defined lower and upper cuts as specified in the text above and in Figs. 6–7. We compared the obtained significant frequencies of our MTM periodograms to results of the modelled La04 astro-nomical solution for the Recent[43] and main frequencies of the precession and obliquity as inferred in[25,56] (Fig. 4a). The best fit of a resulting MTM of the tuned time-series to expected Milankovitch frequencies of the Middle Ordovician was obtained with the dynamic tuning (frequency stabilization) of the time-series by picking the powerful and particularly continuous 17 kyr component of the pre-cession (Supplementary Fig. 10; Supplementary Data 1). MTM periodograms (Fig. 4; Supplementary Fig. 10), evo-FFTs (Fig. 6; Supplementary Figs 10–11) and the COCO procedure (Supplementary Fig. 12) further support our cyclostrati-graphic interpretation and were all produced with the Acycle software[57].

**Precession filter output tuning option**. Another tuning was performed using a large bandwidth precession filter output (0.04–0.1 cycles/cm) and setting up an average duration for this orbital component at 20 kyr according to ref. [25] (Supplementary Data 4). This tuning option logically enhances the power of the 20 kyr and erases any expression of frequency and amplitude modulations of the precession but highlights the presence of a significant periodicity at 413 kyr that matches fairly well the expected 405 kyr component (Fig. 4d). This precession tuning procedure brings up interesting observations. An evo-FFT performed on the time-series tuned with this option shows a dominant and stable 20 kyr component (Supplementary Fig. 9). The 405 kyr component shows up in the expected interval of dominance of this orbital frequency (Supplementary Fig. 9). In addition, a 100 kyr frequency shows high power in a restricted interval at 469.6–470.4 Ma and the transition interval from dominance of the precession to dominance of the 405 kyr at 468.9–469.2 Ma has an interesting expression of a 34 kyr obliquity component that is particularly well-expressed in the evo-FFT, as well as with a significant frequency peak at 167 kyr that may be related to the s3–s6 obliquity amplitude variation cycle (175 kyr at present-day and stable in the past until at least 56.2 Ma[48] although this term is likely to have a different duration in the Ordovician due to the significantly shorter duration of the main obliquity component at that time). It is noteworthy that the 34 kyr obliquity component also appears, albeit with low power, in the upper part of the section in the evo-FFT (Supplementary Fig. 9).

**Frequency stabilization of the 17 kyr precession component**. The best tuning option that we retained was obtained by picking up one of the precession com-ponents that appears particularly continuous and stable from ca. 0.063 to 0.0072 cycles/m (16–14 cm) along the time-series in an evo-FFT and setting it up to 17 kyr, i.e. the duration proposed by Waltham[25] for the P2 component at 468 Ma (Fig. 4a). This procedure that we call frequency stabilization is shown in Supplementary Fig. 10 where we also highlight a likely P1 component of ~20 kyr. This figure highlights significant shifts in frequency of the interpreted precession component as well as potential shifts in frequency of the interpreted 100 kyr and obliquity components (Supplementary Fig. 10d). The thick black curve following P2 is drawn following the points that were picked for reconstructing sedimentation rate variations at given heights in the section (Supplementary Fig. 10e). Accord-ingly, reconstruction of sedimentation rate variations translates into an age model used for tuning the time-series in the time domain in Acycle[57]. This frequency stabilization tuning option provides better results than the precession tuning and 405 kyr tuning options with the power of precession P1 and P2 components faithfully recovered and a duration of 20 for P1 (Supplementary Fig. 11) that matches well what is expected for this component by Waltham[25] (Fig. 4a). Two poorly significant frequencies at 31 and 36 kyr potentially match obliquity components[25,56], and a poorly significant peak at 98 kyr suggests the presence of the short-eccentricity. Moreover, the evo-FFT of the tuned time-series shows that the latter frequency bears several regular bifurcations that typically match ampli-tude modulations of the 100 kyr by the 405 kyr cycle (dotted lines on Supple-mentary Fig. 10b). The 405 kyr component shows a very stable frequency in the evo-FFT of this tuning option with high power at the base of the section and in the upper half (Supplementary Fig. 10b). The evo-FFT of our frequency stabilization tuning option also faithfully replicates the prominent changes in dominance of the precession and 405 kyr components similarly to what is shown in the depth domain in Fig. 6. The frequency stabilization tuning option is our favourite option because, despite tuning on a precession component, this procedure still preserves

frequency modulations of the precession, highlights a very stable 405 kyr component, a weak but likely obliquity component, as well as a weak short-eccentricity component with well-expressed amplitude modulations (Supplementary Figs 10b and 15). A three-dimensional view of a wavelet transform[58] of the time-series tuned with the frequency stabilization option similarly highlights prominent amplitude modulations of the 100 kyr and generally compares well to the expression of a 3D wavelet of the La2004 insolation solution (Supplementary Fig. 15). Note that the amplitude modulations of the precession filter output shown in Fig. 6b show significant frequencies in the short-eccentricity band that are not present in the original lithological rank dataset (Fig. 6a). They are an artefact from the filtering process which transfers original frequency modulations of precession (thickness variations) into power in the resulting filter output.

**COCO validation.** The validity of our interpretation has been further tested by using the COCO approach[59] that investigates the coefficient of correlation between frequency peaks detected by a user in the periodogram of a time-series to the frequencies of orbital targets and evaluates the probability likelihood of a non-orbital influence (null hypothesis) for a range of sedimentation rates. Orbital targets were set in that procedure following Waltham[25] and Svensen et al.[60] to account for the relatively large uncertainty in the known duration of the obliquity. The highest correlation coefficients coincident with the lowest probability for the null hypothesis is obtained for sedimentation rates ranging from 0.75 to 0.85 cm/kyr (Supplementary Fig. 12a, b). The $2\pi$ MTM periodogram of the time-series tuned with a linear sedimentation rate of 0.856 cm/kyr bears frequency peaks that compare well with the orbital targets that we set up in the procedure (Supplementary Fig. 12c, d).

## Data availability
The Supplementary Data 1–6 generated in this study have been deposited in the Electronic Research Data Archive (ERDA) of the University of Copenhagen repository under the following link: https://doi.org/10.17894/ucph.ecb88eb4-7ce5-4f8f-8ee0-305a3a4c179f.

## Code availability
All mathematical treatments have been performed with the freeware Acycle designed by ref. [57] and made publicly available by these authors at https://github.com/mingsongli/acycle.

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

## Acknowledgements

We thank Linda Hinnov (George Mason University) for comments and discussion that helped us improve the results. J.A.R. acknowledges financial support from the Carlsberg Foundation; C.M.Ø.R. and N.T. acknowledge support from Geocenter Denmark projects 2015-5 and 3-2017. Our discussion strongly benefitted from fruitful exchanges with Anders Lindskog (Lund University), David De Vleeschouwer (MARUM, University of Bremen) and Christian Zeeden (Leibniz Institute for Applied Geophysics). This is a contribution to IGCP Project 653: The onset of the Great Ordovician Biodiversification Event. This paper was significantly improved thanks to the thorough comments by Sietske Batenburg and two anonymous reviewers.

## Author contributions

J.A.R. designed the research, conducted fieldwork and sampling, picked and analysed conodont assemblages, and established the lithological rank index. C.M.Ø.R. provided further support to stratigraphic correlations and interpretations and co-designed several illustrations. N.T. analysed the data, designed the illustrations, and drafted the paper with J.A.R. and C.M.Ø.R.

## Competing interests

The authors declare no competing interests.
