## [Peer Review File · Nature Communications]

REVIEWER COMMENTS

Reviewer #1 (Remarks to the Author):

This manuscript presents a high-resolution astrochronology for the Great Ordovician Biodiversification Event (GOBE), which demonstrates a shift in the Earth's orbit beat. It argues that this shift is responsible for the GOBE. Here are four major comments, which may be able to help improve this manuscript no matter whether nature communications publish it or not.

1. Concept and definition of the GOBE

It is now clear that the GOBE is not one, but a number of events rooted in the Cambrian, unfolding during the Ordovician (Rasmussen et al., 2016; Servais and Harper, 2018). Among these events, the main radiation of the GOBE has occurred during the Middle Ordovician Darriwilian Stage. This long event lasts over tens of million years, which is much longer than the time interval covered in this study. These events often display a two-phased rise in diversity with an onset in the late Floian–early Dapingian and a second main spike during the mid-Darriwilian. (Rasmussen et al., 2016).

This manuscript doesn't have a clear definition of the GOBE and its onset. For example, lines 15-17 notes that "astronomically calibrated cyclostratigraphical framework through the onset of the Great Ordovician Biodiversification Event (GOBE)" and "with the start of the radiations" while line 204 shows that "coinciding with the main phase of the GOBE". This manuscript actually focused on the driver(s) of the mid-Ordovician ice age, which is a very brief interval and may be handled by this study. Therefore, the title and content should be refocused.

2. Driving forcing

A recent paper by Schmitz et al. (2019) linked the extra-terrestrial dust associated with an asteroid breakup to the mid-Ordovician ice age again. Alternatively, this paper shows the meteor bombardment to post-date the icehouse by 800 kyr, instead pausing the GOBE 600 kyr after its initiation. Indeed, the latter conclusion has already been reached by Lindskog et al. (2017), who concluded that the Middle Ordovician meteorite bombardment and the GOBE were unrelated. However, it is still important to see that this manuscript proved the finding by Lindskog et al. (2017) using a different, cyclostratigraphic approach.

Fang et al. (2019) reported that the astronomical forcing was responsible for the cooling event during the Darriwilian. They detected a shift in the Earth's orbital beat from eccentricity-dominating interval to obliquity-dominating interval triggering the global cooling and lowstand, as well as a greatly increased richness of graptolites. Alternatively, this manuscript claims that a shift in orbital beat from precession-dominating to eccentricity-dominating was the trigger for global cooling and glaciation. The similarity of these two studies is not discussed thoroughly and the importance of the study by Fang et al. (2019) was underestimated and hidden in the Methods section. This key point should be discussed in the main text, and this may hurt the novelty of this manuscript.

3. Timing & uncertainty

This manuscript used the conodont zonation as the anchor for the floating astronomical time scale. Conodont zones can be diachronous, and the uncertainties of this correlation are not discussed. The influence of the age uncertainty should also be presented.

4. Proxy

Firstly, this paper used proxies of lithofacies and bed thickness, which are subjective proxies and sometimes can have low signal to noise ratios, hampering the recognition of the orbital signals. Secondly, the climate response of these proxies to orbital forcing can vary. For example, Fang et al. (2019) detected strong obliquity forcing from magnetic susceptibility (MS) and gamma ray (GR) data. In comparison, the obliquity

forcing is very weak in the lithofacies and bed thickness series in this study. Therefore, the benefits and disadvantages of proxies should be presented.

References

- Fang, Q., Wu, H., Wang, X., Yang, T., Li, H., Zhang, S., 2019. An astronomically forced cooling event during the Middle Ordovician. *Global and Planetary Change* 173, 96-108.
- Lindskog, A., Costa, M.M., Rasmussen, C.M.Ø., Connelly, J.N., Eriksson, M.E., 2017. Refined Ordovician timescale reveals no link between asteroid breakup and biodiversification. *Nature Communications* 8, 14066.
- Rasmussen, C.M.Ø., Ullmann, C.V., Jakobsen, K.G., Lindskog, A., Hansen, J., Hansen, T., Eriksson, M.E., Dronov, A., Frei, R., Korte, C., Nielsen, A.T., Harper, D.A.T., 2016. Onset of main Phanerozoic marine radiation sparked by emerging Mid Ordovician icehouse. *Scientific reports* 6, 18884.
- Schmitz, B., Farley, K.A., Goderis, S., Heck, P.R., Bergström, S.M., Boschi, S., Claeys, P., Debaille, V., Dronov, A., van Ginneken, M., Harper, D.A.T., Iqbal, F., Friberg, J., Liao, S., Martin, E., Meier, M.M.M., Peucker-Ehrenbrink, B., Soens, B., Wieler, R., Terfelt, F., 2019. An extraterrestrial trigger for the mid-Ordovician ice age: Dust from the breakup of the L-chondrite parent body. *Science Advances* 5, eaax4184.
- Servais, T., Harper, D.A.T., 2018. The Great Ordovician Biodiversification Event (GOBE): definition, concept and duration. *Lethaia* 51, 151-164.

Reviewer #2 (Remarks to the Author):

Dear authors,

I read the manuscript by Rasmussen and co-authors on "A switch in Earth's orbital beat supports glacially-induced Great Ordovician Biodiversification Event" with great interest and congratulate you on the novel findings.

The manuscript shows the careful treatment of lithological data to test for an orbitally-forced origin of the limestone-marl alternations, and establishes an age model that shows that climatic change in the middle Ordovician may have been orbitally triggered and preceded a meteor-bombardment. Some parts of the text could be expanded upon (sometimes just by lifting information from the methods section) but overall, the results are clear and support the interpretation of a switch from precession-dominated sedimentation to a dominant eccentricity pacing of earth's climate. These findings are of interest to the broad scientific community and I recommend publication after minor revisions.

Best wishes,

Sietske Batenburg

Main points:

1. Lithological cycles

The detection of potential orbital forcing in deep time, when the carbonate factory was very different from the present day is intriguing. Would it be possible to incorporate a close up of part of the section (for example the field picture in the lower left-panel of figure 2), with the corresponding rank-coded data next to it, and bundling indicated on the figure and in the data?

In addition, it would be valuable to incorporate the discussion of the origin of the limestone-marl alternations early on, for example at the beginning of the results section, rather than in the methods section. Cycles in carbonate lithologies are typically grouped as either run-off, productivity or dissolution cycles (or a combination). Here, there was delivery of terrigenous material (linked to run-off) and carbonate mud. Does delivery of carbonate mud reflect productivity of the platform, or does it depend mainly on erosion? I am curious if the authors can comment on whether terrigenous or carbonate deposition forms the main variable controlling the lithological alternations.

2. Age control/cycle hierarchy

My second point concerns the interpretation of the cycle hierarchy and the available independent age control, which underlie the astronomical tuning step from the depth domain to the time domain.

Despite the detection of two “groups” of periodicities in the depth domain, there is not a real cycle hierarchy, because the periodicities are observed in different parts of the record. Can you exclude a change in sedimentation rate between these intervals? The authors have probably investigated this, but it would be helpful for the reader to discuss it explicitly, using independent age control. This is a requirement for the discussion of the orbital switch.

The lower interval, dominated by 14–19 cm periodicities, does show a cycle hierarchy, which is particularly convincing in modulation pattern in the evolutive spectrum of the depth series (and supported by further analyses in the time domain).

As the cycle hierarchy is limited (not the full range of eccentricity-modulated precession in one interval) it is particularly important to clearly present the independent age information. Currently, the record is anchored at one point, and independent duration estimates are presented in the text (and in supplementary figure 1, but this figure is not sufficiently annotated to understand). I would like to urge the authors to find a way to present this independent age information more clearly, incorporated in a figure in the main manuscript.

In line 161, the Kinnekulle section should be introduced somewhat. Please include a few sentences on the lithologies and how ages were obtained, and show the position of the site in Figure 1.

3. Data treatment

I appreciate that the raw data and analyses of the raw data are presented clearly. However, many different methods are applied and the reader misses the overview. For example, in the tuning steps, it is difficult to understand why part of the analyses are performed using the 20 kyr cycle of precession, and others using the 17 kyr cycle. Perhaps the addition of an introductory sentence along the lines of: “we applied several time series analyses / astronomical tuning approaches, including 1: xxx, ;, 2: xxx, etc” would be helpful. I’d like to know why you have chosen these approaches, and if the analyses look different if you stick with one of the cycles (for example filtering the 17 kyr component instead of the 20 kyr).

Most time series analyses techniques are sufficiently explained (or referenced) but dynamic tuning might not be familiar to all readers. Define the procedure and describe the criteria for identification of the 17 kyr cycle throughout the dataset (and refer to an existing dynamic tuning approach).

L 408: provide the upper and lower cuts

L 420–424: provide the bandwidths of the filters for the 20 kyr and 405 kyr components.

4. Orbital switch

The words orbital switch sound like a change in the periodicities of the orbital cycles themselves (like from 1.2 Myr to 2.4 Myr long eccentricity cycles). Here, the orbital switch refers to the way the climate system responded to the orbital forcing (possibly triggered by a node in the grand cycles). Perhaps “orbital switch” is not the best term because the orbits don’t switch. If no other term is appropriate, the authors could make the use of the term “orbital switch” to refer to a climatic response more clear. At line 234 it’s worth elaborating a bit on the Cenozoic orbital switches. Also, have a look at Liebrand et al., PNAS, 2017, who do not detect a switch but rather the absence of precession and obliquity signatures in a record dominated by eccentricity forcing. They investigate the record with bispectral analyses, which help explain the transfer of energy from high to low frequencies in the climate system.

The comparison to ice-volume changes is nice. The authors mention times of speciation/radiation that have been linked to nodes in the astronomical solution, perhaps they could highlight some examples from the references they already provide. Also, the text from lines 438–453 in the methods section might be better

placed in the discussion section.

Edits:

Supplementary Figures 1 and 2: the figure captions do not sufficiently explain what is depicted in the figures.

L 14: change farther to further

L 15: cyclostratigraphical to cyclostratigraphic

L 18: move "by 200,00 years" to follow the word "conditions"

L 20 show to shows

L 22: suggest to suggests

L 29: delete comma

L 32: delete second comma

L 34: delete comma

L 37: delete comma, change were to was

L 46: change should correlate to correlates

Fig 1: please indicate the position of other sections mentioned in the manuscript, such as Kinnekulle.

L 55: change occurring to that occurred

L 80: change which express this lithology to which expresses these lithologies

L 114: change show to shows

Fig 5: Please improve the figure caption. For example, the evolutive spectra are not mentioned, nor is the pink triangle, which currently suggests that a depth interval of >1 m corresponds to an instantaneous moment in time. Why are some pink boxes more transparent than others, and why does one overlie the astrochronology curve? The title "astrochronology" is misleading, use something like "Precession (... cm bandpass-filter)"

L 126: change support to supports

L 127: "dismiss all possible ambiguities sounds vague, please rephrase what you are excluding exactly

L 138: add periodicity after kyr

L 143: delete the before eye, change on to in

L 149: add frequency after precession

L 150: add periodicity after 405 kyr

L 156: add cycle after precession

L 157: change that to those

L 163: delete the before precession

Figure 7: the golden spike symbol is not appropriate

L 202: add cooling before trends to make the sentence more specific

L 208: remind readers what LCPB is

Fig 8: the sea level curves need a scale, or at least an indication of which way is higher and which way is lower. The yellow nail in the stratigraphy column suggests a GSSP "golden spike", please use a different symbol

L 214: change occur to occurs

L 228: "neither ..., nor" should be "either..., or", or you can rephrase by replacing "in" by a comma, and adding "in" after neither and after nor.

L 237: change relate to be related to

L 260: move strongly to before influenced

L 269: add an s after change

L 271: change lead to led

L 296: Define range-through and how this is different from "range"

L 357: change was to were

L 374: the word neat is confusing, please rephrase

L 416: change supporting further to further support

L 427: move "logically" to before "shows" or replace by ", as expected,"

L 428: nicely is not appropriate

L 487: add "that" before "are"

L 489: delete "the"

Reviewer #3 (Remarks to the Author):

Dear Authors,

with great interest, I have read your manuscript thoroughly. You present a lithology rank depth series from the Ordovician of Scandinavia of ~320 m, which corresponds to ~4Ma. You apply cyclostratigraphic analyses, anchor your dataset to a U/Pb age, and derive an absolute time scale for the record. Finally, you use this record to discuss the possible rivers in the orbital imprint, discuss possible causes of the increase in biodiversity and glaciation, and discuss general causes for glaciations and shifts in orbital imprints.

Your findings are relevant in Earth Science and are part of a recent debate on the cause of the Ordovician diversity increase in conodonts. You present an excellent dataset, and the patterns are indeed indicative of your interpretation, which I do not doubt (but I would prefer a different line of argumentation, see below). Your data may be very useful for further follow-up studies. The statistical methods are applied in a useful way, and are discussed in a fair manner (not overstating, but to the point). With the data at hand, I your work looks reproducible.

Your cyclostratigraphic analysis is done thoroughly using both spectral- and amplitude information. The main issue is in my opinion that different cyclicity is present in two parts of the record, but you treat the record as one. It is in my opinion necessary to discuss (and exclude) the possibility of changing sedimentation rate as cause for your findings.

While you present your findings in a structured way, I see space for improvement at several places.

You submit a manuscript to NatureCommunications, a high-impact journal. I would expect some quantitative data. While I agree with the value of a qualitative lithological rank system which you provide, I also think that quantitative data – or a very detailed photographic documentation are needed for publication in such a high-impact journal. While using a depth rank system is not generally a problem and this has been shown to be very powerful data in the past, I'm unsure if this allows for publication in a high-ranking journal as Nature Communications. Personally, I would expect at least some data as 'proof-of-concept' – where e.g. carbonate content is plotted with your rank system for a metre or several.

Generally, I miss an uncertainty assessment/discussion. You give ages with a resolution of 100 ka – is this realistic here? Given that you place question marks beside the interpretation of precession cycles in Fig. 5, this seems unreasonable. This may go in the paragraph starting in line 222.

I would prefer to see an evolutive spectrum (or wavelet) before the power spectra. In this case the power spectra for the full interval may be a problem in my opinion. While you find different cyclicity in different depths intervals (Fig. 5), you show spectra for the full interval. The obvious and unanswered question is if your interpretation may be an artefact of changing sedimentation rate at ~2000cm? While I agree with your interpretation,

Several sentences are unnecessary complicated and hard to read – I ask you to reduce the amount of side-sentences and commas in some sentences (e.g. lines 213ff).

The final part about general changes in Earth's response to climate forcing seems to miss a red line – I'm unsure where you try to bring the reader in the last ~50 lines of text. While I follow all your sentences, their overall message is not obvious.

Some more detailed feedback:

Line 125ff: why can you exclude ambiguities?

Fig. 2: please mark cycle bundles in at least one image

Fig3: which information is directly from this outcrop, which not? Please make sure that this is clear from Figure and caption. I'm unsure if this needs to be part of the manuscript, it may also go to Supplements in my opinion.

Table 1: how is the correlation to carbonate data done, where is the data?

Fig5: the lithology rank and the lithology seem offset a bit, especially at ~2570cm

Fig. 8 needs abscissas for all datasets, and arrows how the data are related to sea level (where is rising/falling?). The yellow line is hardly visible in print.

REVIEWER COMMENTS

Reviewer #1 (Remarks to the Author):

1.1. Concept and definition of the GOBE

It is now clear that the GOBE is not one, but a number of events rooted in the Cambrian, unfolding during the Ordovician (Rasmussen et al., 2016; Servais and Harper, 2018).

We do not agree with this statement and the reference to Rasmussen et al. (2016) is actually out of context here. There is a semantic discussion ongoing based on the wording of the original description of the GOBE made by Webby et al. 2004. For this reason, Servais et al. (2016; 2021), Servais & Harper (2018) and Harper et al. (2020) argue that the GOBE should be viewed as a prolonged phase of radiation spanning the entire Ordovician and with its roots in the late Cambrian. However, they have presented no new data to back up these claims, except for arguing for their case through an analysis based on a temporal binning which is even cruder than the one presented by Webby et al. based on their original description of the GOBE. One has to remember, though, that Webby et al 2004 described the GOBE based on the data at hand at the time of their study, which is almost 20 years ago now and with a much less detailed record as compared to what is available today.

Opposed to this view stands a bulk of new research on the timing of the GOBE across various geographical regions and across different clades that have come out since Webby et al. 2004 first coined the term of “GOBE” (see, for instance, Rasmussen et al., 2007; Harper et al., 2013; Adrain et al 2013; Trubovitz & Stigall 2016; Hints et al., 2018; Cole 2018; Colmenar & Rasmussen 2018; Ernst 2018; Rasmussen et al., 2019; Kröger et al., 2019 and many more). Interestingly, the original Sepkoski dataset show the same trends as these new studies (Sepkoski 1979; 1981; 1995; Sepkoski & Sheehan 1983).

All these studies have been exhaustively reviewed by the recent paper of Stigall et al. (2019) who demonstrate quite clearly that the “GOBE” – as the name implies – should actually be viewed as **a Middle Ordovician ‘event’**, and not as a protracted phase spanning 40 million years. Some clades show a gradual rise earlier in the Ordovician and some originate during the late Cambrian. But neither of these can be characterized as anything else than background speciation, something that basically occurs throughout the Phanerozoic. We do recognize, however, that in South China (or, more specifically in peri-Gondwana), a pre-GOBE radiation commenced during the Early Ordovician and this may well have sparked the GOBE itself (see Zhan & Harper, 2006; Rong et al., 2007; Fan et al., 2020). But, this regional radiation of South China cannot be deemed a global event and does not question the interpretation by Stigall et al. (2019). The recent review of taxonomic data by Stigall et al. (2019) strongly supports a restricted concept of the GOBE to a Middle Ordovician event only. We therefore stand with our statement which is fully justified by the literature.

Among these events, the main radiation of the GOBE has occurred during the Middle Ordovician Darriwilian Stage. This long event lasts over tens of million years, which is much longer than the time interval covered in this study. These events often display a two-phased rise in diversity with an onset in the late Floian–early Dapingian and a second main spike during the mid-Darriwilian.

(Rasmussen et al., 2016).

Again, our response here is the same as above. We fundamentally disagree with this view and the reference provided does not even support it. The reference to Rasmussen et al. 2016 (CMØ Rasmussen is one of the 3 authors of the present manuscript) is invalid to support the reviewer's statement as this paper specifically denotes that the two-phased rise refers to the fact that some planktonic groups, notably the phytoplankton, rise in diversity during the early Ordovician. Further, there is a radiation across clades in the same interval in South China. However, this early phase cannot be termed a global speciation phase. In addition, it is not relevant for the current study **as we state that we focus on the initiation of the main phase of the GOBE which is well agreed upon by various authors as cited above to commence during the earliest Darriwilian Age. And as we do indeed try to understand what possibly sparked this main phase of diversity rise, our study looks exactly at the right time interval.**

This manuscript doesn't have a clear definition of the GOBE and its onset. For example, lines 15-17 notes that "astronomically calibrated cyclostratigraphical framework through the onset of the Great Ordovician Biodiversification Event (GOBE)" and "with the start of the radiations" while line 204 shows that "coinciding with the main phase of the GOBE". This manuscript actually focused on the driver(s) of the mid-Ordovician ice age, which is a very brief interval and may be handled by this study. Therefore, the title and content should be refocused.

-Again, we believe that this is very much a view of semantics and that, in our opinion, it is not correct. We have a very clear definition as we strictly follow the well-founded views of Stigall et al (2019) which is similar to our definition. Further, as the reviewer points out, Fang et al (2019) also finds evidence for a Darriwilian icehouse. However, as this is later in the Darriwilian (see below and see also Cherns et al., 2013), this supports the notion of the onset of a considerably prolonged icehouse climate. The reviewer's statement that the Middle Ordovician icehouse '*is a very brief interval*' is incorrect as icehouse conditions lasted at least 6–7 million years (Trotter et al., 2008; Rasmussen et al., 2009; Turner et al., 2012; Cherns et al., 2013; Dabard et al., 2015; Rasmussen et al., 2016; Fang et al., 2019), and arguably continuing well into the Sandbian Age (Vandenbroucke et al., 2010; Saltzman & Young, 2005).

We do, however, acknowledge the reviewer's point of view and for this reason, we have decided to be more precise in what we consider and define as GOBE in our paper, following Stigall et al. 2019. **See Lines 340-345.**

1.2. Driving forcing

A recent paper by Schmitz et al. (2019) linked the extra-terrestrial dust associated with an asteroid breakup to the mid-Ordovician ice age again.

It is not correct to state that Schmitz et al. (2019) demonstrated this link twice. The first paper of Schmitz et al. (2008) actually linked the GOBE to the asteroid breakup (LCPB) but this was before anyone had discussed any link between the GOBE and cooling climate (see Trotter et al., 2008; Rasmussen et al., 2009). Schmitz et al. (2019) was the first to then suggest that the Middle Ordovician icehouse was induced by the LCPB.

Alternatively, this paper shows the meteor bombardment to post-date the icehouse by 800 kyr, instead pausing the GOBE 600 kyr after its initiation. Indeed, the latter conclusion has already been

reached by Lindskog et al. (2017), who concluded that the Middle Ordovician meteorite bombardment and the GOBE were unrelated.

How could the latter conclusion be reached by Lindskog et al. (2017) when Schmitz et al. (2019) made their claim two years later? Actually Lindskog et al. 2017 showed a new synthetic framework that supports that the LCPB postdated the onset of the GOBE but there was no astrochronological control on their geochronometer and the timing between the two events is not evaluated in the latter study. The reception of the paper by Schmitz et al. (2019), which had a huge mediatic impact, quite clearly shows the need to reassess the timing and sequence of the three geological events (glaciation, radiation and LCPB) to better understand eventual causal relationships between them. Without the new constraints that we bring in our paper, Lindskog et al. (2017) could not infer accurate temporal constraints on the onset of the GOBE in relation to the LCPB.

In addition, Lindskog et al. (2017) had no information (nor did they have any discussion at all) on the onset of the Middle Ordovician icehouse. As the hypothesis of a climate-link to the LCPB was not brought forward until two years later by Schmitz et al. (2019) the Lindskog paper could not possibly have ‘reached that conclusion’.

Lastly, and this is one of the main findings of our current study, our study is the first one to bring forward that the LCPB actually acted as a brake on the unfolding GOBE and not as a spark. This is certainly not a conclusion that had been ‘reached’ already by Lindskog et al. (2017) and we believe that it participates to make our study of fundamental importance.

However, it is still important to see that this manuscript proved the finding by Lindskog et al. (2017) using a different, cyclostratigraphic approach.

As answered above, the Lindskog paper is only relevant in the context of linking the GOBE and the LCPB. However, Schmitz et al. (2019) still claim a link between the two via the possible effect of the LCPB as sparking glaciation. Another major finding here – besides constraining, for the first time, an absolute temporal perspective of a major biological event nearly half a billion years ago – is that we show that the climatic cooling *also* preceded the LCPB.

Fang et al. (2019) reported that the astronomical forcing was responsible for the cooling event during the Darriwilian. They detected a shift in the Earth’s orbital beat from eccentricity-dominating interval to obliquity-dominating interval triggering the global cooling and lowstand, as well as a greatly increased richness of graptolites. Alternatively, this manuscript claims that a shift in orbital beat from precession-dominating to eccentricity-dominating was the trigger for global cooling and glaciation. The similarity of these two studies is not discussed thoroughly and the importance of the study by Fang et al. (2019) was underestimated and hidden in the Methods section. This key point should be discussed in the main text, and this may hurt the novelty of this manuscript.

This is the third statement by the reviewer aiming at toning down, or undermining the importance of our study and again, this statement is not fair and also incorrect.

The Fang et al. 2019 paper is not “hidden” voluntarily in our Methods section. It is relevant, as stated in our Methods section but it is not the exact same timing as our study and it is also not the same results. Fang et al. (2019) deal with an interval that loosely spans the Darriwilian radiation but without any actual clear correlation to the traditional biozones of the Darriwilian. Their paper does

not even show how their section correlates to other sections in China where the bulk of the radiation in graptolite and other invertebrate groups occurs. Their synthetic figure 8 is not supported by any clear prior figures that actually demonstrate the correlation of various processes, proxies and temperature changes that they present, all these links are just loosely drawn along the same time line. The potential cooling that they discuss as potential response to an orbital switch from eccentricity to obliquity occurs 4 Myr before the base Sandbian but with no timing from the top Dapingian. Therefore, their data, as opposed to the current study, are not geochronologically constrained and only operate a ‘floating astrochronology’ from the Base Sandbian, with no clear link to the GOBE. They have no constraints on the lower boundary of the Darriwilian Stage, nor any clear constraints to the timing of peculiar invertebrate groups that show a radiation such as conodonts and brachiopods in our study. Therefore, it is clear that their paper was not relevant enough to discuss in the context of ours. Due to the loose anchoring of Fang et al. (2019) data, it is actually not even possible to even correlate our results to theirs with certainty. Therefore, we stand with our claim that our study is novel and fundamental. Regarding our discussion of Fang et al. (2019), **see lines 443-446**. We see no reason to discuss further the Fang et al. study in more detail than it already is in the paper.

1.3. Timing & uncertainty

This manuscript used the conodont zonation as the anchor for the floating astronomical time scale. Conodont zones can be diachronous, and the uncertainties of this correlation are not discussed. The influence of the age uncertainty should also be presented.

We do acknowledge that conodont zones may be diachronous, such as any biozones, e.g. trilobite, brachiopod or graptolite zones. A great advantage with regard to conodonts, however, is that several deeper-water taxa apparently had a pelagic, free-swimming life style, which allowed them to migrate fast between continents and make very precise correlations between platform margin sections worldwide (e.g. Rasmussen 1998). Yet, there is no reason to think that conodont zones would be more diachronous than any other fossil group, and conodont zonation has had immense success in correlating shelf carbonate sections throughout Baltica as well as between continents (Albanesi & Bergström 2010; Bergström 1971, 1986; Bergström & Ferretti 2017; Bergström & Löfgren 2009; Bergström et al. 2009; Löfgren 1978; Stouge 1984; Stouge et al. 2020; Rasmussen 2001; Rasmussen & Stouge 2018; Zhang 1998). The correlation of conodont zones to trilobite and graptolite zones in the Lindskog study is supported by decades of studies on the stratigraphy of the Middle Ordovician. To go one step further in the line of the reviewer’s argument would suppose to ignore the published stratigraphic scheme of Lindskog, to ignore the exact same correlation of trilobite and conodont zone by Schmitz et al., and then cast doubt over the correlation between conodont zones and the major radiation of brachiopods as shown in Figure 8. Even with such an argumentation that would widely ignore these published schemes, it would not take away the fact that we also find a very clear radiation of conodonts in our study which delineates the onset of the GOBE’s main phase, and, non-surprisingly, coincides precisely with that of brachiopods when using well-established stratigraphic schemes. Regarding uncertainty, it is now widely dealt with in our new version as demonstrated further in response to other reviewers’ comments.

1.4. Proxy

Firstly, this paper used proxies of lithofacies and bed thickness, which are subjective proxies and

sometimes can have low signal to noise ratios, hampering the recognition of the orbital signals. Secondly, the climate response of these proxies to orbital forcing can vary. For example, Fang et al. (2019) detected strong obliquity forcing from magnetic susceptibility (MS) and gamma ray (GR) data. In comparison, the obliquity forcing is very weak in the lithofacies and bed thickness series in this study. Therefore, the benefits and disadvantages of proxies should be presented.

Good point. However, we point out here that the excellent paper by Olsen et al. 2019, PNAS, “Mapping Solar System chaos with the Geological Orrery” relies precisely on a cyclostratigraphic analysis and astronomical calibration of lithofacies rank data and bed thickness. The advantage of a lithofacies signal is its sensitivity to frequency modulations. The division in various lithofacies of same values perhaps hampers the recognition of the full spectrum and amplitude of orbital components but it allows to assess variations in thickness of couplets, which, if controlled by precession, should show frequency modulations by the eccentricity, and this is exactly the demonstration that we make here in our paper. **We now address this point better in lines 478-485.**

Reviewer #2 (Remarks to the Author):

Main points:

2.1. Lithological cycles

The detection of potential orbital forcing in deep time, when the carbonate factory was very different from the present day is intriguing. Would it be possible to incorporate a close up of part of the section (for example the field picture in the lower left-panel of figure 2), with the corresponding rank-coded data next to it, and bundling indicated on the figure and in the data?

We have now modified Figure 2 with the addition of a figure showing precisely what was asked by the reviewer with explanations of lithofacies and colour coding as in the modified Table 1. We also added a figure (Figure 5) showing a new analysis of high-resolution grey-level data extracted from the best picture we had of one of the intervals situated at 2500 to 2800 cm, which supports the identification of not only the precession but also short-eccentricity cycles, which could not be identified in our lithology rank data. This new analysis supports our previous interpretation. **In addition, we have also incorporated close-up photographs with explanations of lithofacies from various parts of the section in the Supplementary Information, which demonstrate the overall cyclic bedding pattern. See lines 167-181 and see our revised appendix with supplementary figures 9 to 15**

In addition, it would be valuable to incorporate the discussion of the origin of the limestone-marl alternations early on, for example at the beginning of the results section, rather than in the methods section. Cycles in carbonate lithologies are typically grouped as either run-off, productivity or dissolution cycles (or a combination). Here, there was delivery of terrigenous material (linked to run-off) and carbonate mud. Does delivery of carbonate mud reflect productivity of the platform, or does it depend mainly on erosion? I am curious if the authors can comment on whether terrigenous or carbonate deposition forms the main variable controlling the lithological alternations.

Production versus dilution cycles or even dissolution cycles are typical controls explaining variations in CaCO_3 in sediments and it is not easy to decipher which process is the dominant

control. In the absence of any geochemical data, we cannot comment on this. But we agree with the reviewer to incorporate the discussion on the origin of the limestone-marl alternations at the beginning of the results section. **This is now done in the new version lines 99-136.**

2.2. Age control/cycle hierarchy

My second point concerns the interpretation of the cycle hierarchy and the available independent age control, which underlie the astronomical tuning step from the depth domain to the time domain.

Despite the detection of two “groups” of periodicities in the depth domain, there is not a real cycle hierarchy, because the periodicities are observed in different parts of the record. Can you exclude a change in sedimentation rate between these intervals? The authors have probably investigated this, but it would be helpful for the reader to discuss it explicitly, using independent age control. This is a requirement for the discussion of the orbital switch.

Our age control here is the conodont biostratigraphy which appears in Lindskog et al. 2017 synthetic stratigraphic framework, whose study actually provides clear geochronological age constraints on our studied interval. According to the latter study, our interval, which spans close to the base of the *P. originalis* Zone to lowermost *E. pseudoplanus* Zone should be restricted to an interval comprised between 467 and 470.8 Ma, i.e. with an average sedimentation rate of 0.85 cm/kyr. The results of our study, once anchored to the base of the *Y. crassus* Zone at 467.5 Ma points to an interval comprised between 467.26 and 471.08 Ma, with an average sedimentation rate of 0.85 cm/kyr. This is quite remarkably similar to the Lindskog study. Changes in sedimentation rates are observed and delineated in various detailed cyclostratigraphic analysis in our appendix. Moreover, the new documentation of the grey level signal analysed in the upper part of the section where we could not depict well the precession but only the 405 kyr now further supports the coincident presence of not only two groups of periodicities (precession and 405 kyr but also of the 100 kyr).

We have now added two more anchors derived from the study of Lindskog et al. 2017 at 468.42 Ma for the base of the *L. variabilis* Zone and at 470.1 Ma for the base of the *B. norrlandicus* Zone in the newest version of Figure 6 (see the two green circles with CRE ages) in order to show the potential match and discrepancies between their study and ours. **See also our text lines 315-317.**

The lower interval, dominated by 14–19 cm periodicities, does show a cycle hierarchy, which is particularly convincing in modulation pattern in the evolutive spectrum of the depth series (and supported by further analyses in the time domain).

As the cycle hierarchy is limited (not the full range of eccentricity-modulated precession in one interval) it is particularly important to clearly present the independent age information. Currently, the record is anchored at one point, and independent duration estimates are presented in the text (and in supplementary figure 1, but this figure is not sufficiently annotated to understand). I would like to urge the authors to find a way to present this independent age information more clearly, incorporated in a figure in the main manuscript.

Very good point, we now indicate the independent age constraints given by Lindskog et al. 2017 for the base of the *L. variabilis* Zone (CRE age) and for the base of the *B. norrlandicus* Zone on figure 6 (CRE age) which strongly support our interpretation. We also refer to Figure 8 where we compare

the independent original time-scale of Lindskog to our astronomical time scale. See lines 303-317.

In line 161, the Kinnekulle section should be introduced somewhat. Please include a few sentences on the lithologies and how ages were obtained, and show the position of the site in Figure 1.

Good point. We have now added a sentence introducing this section better and explaining how it is anchored. We have also added the references Lindskog and Eriksson (2017) and Jaanusson (1973) and have included the locality Kinnekulle in Figure 1. Lines 266–301.

2.3. Data treatment

I appreciate that the raw data and analyses of the raw data are presented clearly. However, many different methods are applied and the reader misses the overview. For example, in the tuning steps, it is difficult to understand why part of the analyses are performed using the 20 kyr cycle of precession, and others using the 17 kyr cycle. Perhaps the addition of an introductory sentence along the lines of: “we applied several time series analyses / astronomical tuning approaches, including 1: xxx, ;, 2: xxx, etc” would be helpful. I’d like to know why you have chosen these approaches, and if the analyses look different if you stick with one of the cycles (for example filtering the 17 kyr component instead of the 20 kyr).

Done. See line 619 justified by Waltham (2015) for the resultant average duration of the precession and not focusing on one particular frequency of the whole precession window. See lines 652 to 656 for the dynamic tuning on the P2 component of the precession at 17 kyr. Filtering out solely the 17 kyr component with a narrow frequency window rather than the complete precession window with a wider frequency band logically results into suppressing frequency and amplitude modulations and is therefore not useful here. See lines 612-614 for justifying the use of several approaches to further support our interpretation.

Most time series analyses techniques are sufficiently explained (or referenced) but dynamic tuning might not be familiar to all readers. Define the procedure and describe the criteria for identification of the 17 kyr cycle throughout the dataset (and refer to an existing dynamic tuning approach).

We agree. Good point. As explained above, we do this now in the manuscript. See lines 653 to 658

L 408: provide the upper and lower cuts

They were already given in Figures 6–7 and we have also precised them now earlier in the text so we simply added at the end of our sentence “as precised in the text above and in Figs 6–7”. See line 232 and line 618 for the precession, see line 237 for the 405 kyr

L 420–424: provide the bandwidths of the filters for the 20 kyr and 405 kyr components.

Done. See response above

2.4. Orbital switch

The words orbital switch sound like a change in the periodicities of the orbital cycles themselves (like from 1.2 Myr to 2.4 Myr long eccentricity cycles). Here, the orbital switch refers to the way the climate system responded to the orbital forcing (possibly triggered by a node in the grand cycles). Perhaps “orbital switch” is not the best term because the orbits don’t switch. If no other term is appropriate, the authors could make the use of the term “orbital switch” to refer to a climatic response more clear. At line 234 it’s worth elaborating a bit on the Cenozoic orbital switches. Also, have a look at Liebrand et al., PNAS, 2017, who do not detect a switch but rather the absence of precession and obliquity signatures in a record dominated by eccentricity forcing. They investigate the record with bispectral analyses, which help explain the transfer of energy from high to low frequencies in the climate system.

Thank you so much for this reference which greatly helps our case indeed. We now mention this paper too in our revised version. Also, following the reviewer’s advice, we avoid the use of the term “orbital switch” for our own example which the reviewer suggests being improper and rather use “shift” to refer to the change from precession to 405 kyr dominance. We also subsequently modified the title to: **“Middle Ordovician astrochronology decouples asteroid breakup from glacially-induced biotic radiations”**. We added mention to Liebrand et al. 2017 which supports very much our hypothesis. **See lines 391-400**

The comparison to ice-volume changes is nice. The authors mention times of speciation/radiation that have been linked to nodes in the astronomical solution, perhaps they could highlight some examples from the references they already provide.

This was already done in the text and we would rather not expand this part as the correlation of the results obtained in the study by Crampton et al. 2018 that spans a very long interval to ours could only be done with large uncertainties. **See reference to Crampton et al. lines 415-417**

Also, the text from lines 438-453 in the methods section might be better placed in the discussion section.

Done. This has been replaced in the discussion accordingly lines 426 to 436.

Reviewer #3 (Remarks to the Author):

While you present your findings in a structured way, I see space for improvement at several places. You submit a manuscript to NatureCommunications, a high-impact journal. I would expect some quantitative data.

While I agree with the value of a qualitative lithological rank system which you provide, I also think that

3.1. Quantitative data – or a very detailed photographic documentation are needed

for publication in such a high-impact journal. While using a depth rank system is not generally a problem and this has been shown to be very powerful data in the past, I’m unsure if this allows for publication in a high-ranking journal as Nature Communications. Personally, I would expect at least some data as 'proof-of-concept' – where e.g. carbonate content is plotted with your rank system for a metre or several.

Done, we now present the result of the analysis of a high-resolution grey-level signal produced from one of our best pictures of the section, the analysis supports our previous interpretation (**See our new figure 5 and text associated to this new figure lines 168 to 181**).

Moreover, we have added detailed photographs from the lower, middle and upper parts of the analysed section with named lithofacies, which document that more pure limestone beds alternate with more siliciclastic-rich limestone beds in a cyclic pattern throughout the measured section. **All new photos with an overlay of interpreted facies with colour coding** as given in **Table 1** have been added to the **Supplementary information file (suppl figs 8 to 12)**. Moreover, we also provide description of a number of **thin sections** for various facies in the Supplement (suppl figs 13-15). We do not however directly link carbonate content data to our lithofacies as we cannot provide systematic CaCO₃ measurements of our lithofacies. We think, however, that the addition of a high-resolution digital grey level signal + documentation of our lithofacies by macro-photographs + microfacies photographs is enough to justify our study.

Generally,

3.2. I miss an uncertainty assessment/discussion.

You give ages with a resolution of 100 ka – is this realistic here? Given that you place question marks beside the interpretation of precession cycles in Fig. 5, this seems unreasonable. This may go in the paragraph starting in line 222.

Providing uncertainties is an especially tedious task with cyclostratigraphy. However, we have now taken this comment into account and added a full account of uncertainties, explaining the differences obtained in the various kinds of tuning approaches (3990+/-170 kyr) but also expanding these uncertainties based on the results of our COCO procedure that provides a range of potential sedimentation rates from 0.7 to 0.85 cm/kyr, thus pointing to maximum duration of ~4642 kyr versus minimum of 3823 kyr which we favor due to being in line with our favored dynamic tuning. **See lines 253-263.**

Accordingly, we now provide a full account of uncertainties for the base of the various conodont zones: “Considering the latter tuning approach and taking the base *Y. crassus* as our radiometric anchor, we estimate a maximum cyclostratigraphic uncertainty of +0.24 Ma for the base of *L. variabilis*, +0.56 Ma for the base of *B. norrlandicus*, +0.70 for the base of *M. parva* and +0.76 Ma for the base of *P. originalis*”. **See lines...253-263**

I would prefer to see an evolutive spectrum (or wavelet) before the power spectra. In this case the power spectra for the full interval may be a problem in my opinion. While you find different cyclicity in different depths intervals (Fig. 5), you show spectra for the full interval. The obvious and unanswered question is if your interpretation may be an artefact of changing sedimentation rate at ~2000cm?

This comment is essentially similar to the previous comment regarding observation of precession in our lower part of the section and 405 kyr in the upper part of the section. We think that we now respond to this issue by the results of our new analysis of grey levels (**new Figure 5**). Moreover, our most important figure (**Figure 6**) does show evolutive power spectra (**two distinct spectra focusing on different frequency windows to highlight as best as possible the power of the two main components**). Moreover, we also show in our Supplement two distinct MTM power spectra of the lithology rank signal below and above 2000 cm (**Supplementary Figure 3**).

While I agree with your interpretation, several sentences are unnecessary complicated and hard to read – I ask you to reduce the amount of side-sentences and commas in some sentences (e.g. lines 213ff).

Duly noted and corrected. This part has been rewritten, sentences have been simplified and cut. See lines 368-378

3.3. The final part about general changes in Earth's response to climate forcing seems to miss a red line – Im unsure where you try to bring the reader in the last ~50 lines of text.

While I follow all your sentences, their overall message is not obvious.

Ok. we have expanded this part slightly following the recommendations of the previous reviewer to include in this part of the discussion a part that was in the methods section. Subsequently, this part of the discussion has also been rewritten and we hope that our final message is clear: “the energy transfer from precession to 405 kyr eccentricity represents a response of the Earth's climatic system to a significant increase in ice volume, favored by a peculiar orbital configuration such as a node of a 1.2 and/or 2.4 myr grand cycles”. See lines 391-456

Some more detailed feedback:

Line 125ff: why can you exclude ambiguities?

The idea we wanted to express was that if the periodicity was not from an orbital origin, then we would not find frequency modulations with ~1:5 cycle bundling. However, this formulation of “dismiss all possible ambiguities” may be poorly expressed and we therefore decided to delete it completely.

Fig. 2: please mark cycle bundles in at least one image

It is not easy to depict bundles with absolute accuracy visually on the pictures. As explained in the paper, the 100 kyr cannot be clearly depicted from the original signal and only shows up while looking at the signal of the cycle thickness (the Frequency Modulation analysis). Therefore, the bundles that we mention are the ones depicted in the cycle thickness time-series of Figure 7. These bundles are depicted by the 100 kyr filter output in this figure. However, we now also show the potential expression of the 100 kyr in the grey level signal of Figure 5 which corresponds to the same image as the lower-right picture of Figure 2.

Fig3: which information is directly from this outcrop, which not? Please make sure that this is clear form [Figure](#) and caption. Im unsure if this needs to be part of the manuscript, it may also go to Supplements in my opinion.

Figure 3 presents the complete section that is continuously exposed at Steinsodden. All informations derive from the section, except the trilobite zones which are drawn by correlation to other sections in Baltica. This is now precised in the figure caption. In our opinion, the illustration of this figure is important for providing the necessary stratigraphic context.

Table 1: how is the correlation to carbonate data done, where is the data?

This is a very good point. The CaCO₃ data shown in Table 1 were based on average estimates from only very few samples across the Stein Formation and originating from the monograph of Rasmussen 2001. The method applied ([Ca] by atomic absorption spectrophotometer and recalculated to %CaCO₃ by assuming no influence of calcic plagioclase and other Ca-rich minerals)

bears large uncertainties and the data are too few to truly represent reliable estimates. Therefore, we decided to completely remove this information. However, the CaCO₃ content proposed for the lithofacies were just rough estimates to express that the ranking essentially represents various amounts of clay. We agree that this should have been explained more clearly in the text or dropped. The present lithofacies distinctions are based on visual, lithological differences between the alternating beds because these are easily recognizable in the field. We believe that it is important to support the lithofacies by other means and have therefore added **new close-up photographs with lithofacies indicated in figure 2 and in the Supplementary figures 8-12. Moreover, thin sections stained with alizarin red to indicate the relative CaCO₃ content have been produced from few, selected levels representing different lithofacies (see Supplementary figures 13-15).** It is evident from both field observations and thin sections, that the pure limestone lithofacies contain more CaCO₃ than the more clay-rich, marly limestone facies (deeper red colour), which support the chosen lithofacies division shown in Table 1. Atomic absorption spectrophotometer % CaCO₃ values from four of the eight shown thin sections (Supplementary Information Figures 13A, 14A, 14B, 15A) were available and are mentioned in the respective figure captions.

Fig5: the lithology rank and the lithology seem offset a bit, especially at ~2570cm
Corrected. See new version, figure 6 now

Fig. 8 needs abscissas for all datasets, and arrows how the data are related to sea level (where is rising/falling?). The yellow line is hardly visible in print.
Corrected. See new version, figure 9 now.

Minor Edits asked by Reviewer 2:

Supplementary Figures 1 and 2: the figure captions do not sufficiently explain what is depicted in the figures.

The authors: corrected.

L 14: change farther to further

The authors: corrected.

L 15: cyclostratigraphical to cyclostratigraphic

The authors: corrected.

L 18: move “by 200,00 years” to follow the word “conditions”

The authors: corrected.

L 20 show to shows

The authors: corrected.

L 22: suggest to suggests

The authors: corrected.

L 29: delete comma

The authors: corrected.

L 32: delete second comma

The authors: corrected.

L 34: delete comma

The authors: corrected.

L 37: delete comma, change were to was

The authors: corrected.

L 46: change should correlate to correlates

The authors: corrected.

Fig 1: please indicate the position of other sections mentioned in the manuscript, such as Kinnekulle.

The authors: done.

L 55: change occurring to that occurred

The authors: corrected.

L 80: change which express this lithology to which expresses these lithologies

The authors: corrected.

L 114: change show to shows

The authors: corrected.

Fig 5: Please improve the figure caption. For example, the evolutive spectra are not mentioned, nor is the pink triangle, which currently suggests that a depth interval of >1 m corresponds to an instantaneous moment in time. Why are some pink boxes more transparent than others, and why does one overlie the astrochronology curve? The title “astrochronology” is misleading, use something like “Precession (... cm bandpass-filter)”

The authors: done.

L 126: change support to supports

The authors: corrected.

L 127: “dismiss all possible ambiguities sounds vague, please rephrase what you are excluding Exactly

The authors: deleted and then reformulated.

L 138: add periodicity after kyr

The authors: corrected.

L 143: delete the before eye, change on to in

The authors: corrected.

L 149: add frequency after precession

The authors: corrected.

L 150: add periodicity after 405 kyr
The authors: corrected.

L 156: add cycle after precession
The authors: corrected.

L 157: change that to those
The authors: corrected.

L 163: delete the before precession
The authors: corrected.

Figure 7: the golden spike symbol is not appropriate
The authors: corrected. The nail is now grey to avoid confusion.

L 202: add cooling before trends to make the sentence more specific
The authors: corrected.

L 208: remind readers what LCPB is
The authors: corrected.

Fig 8: the sea level curves need a scale, or at least an indication of which way is higher and which way is lower. The yellow nail in the stratigraphy column suggests a GSSP “golden spike”, please use a different symbol
The authors: corrected. The nail is now grey to avoid confusion.

L 214: change occur to occurs
The authors: corrected.

L 228” “neither ..., nor” should be “either..., or”, or you can rephrase by replacing “in” by a comma,
The authors: corrected.

and adding “in” after neither and after nor.
The authors: corrected.

L 237: change relate to be related to
The authors: corrected.

L 260: move strongly to before influenced
The authors: corrected.

L 269: add an s after change
The authors: corrected.

L 271: change lead to led
The authors: corrected.

L 296: Define range-through and how this is different from “range”
The authors: corrected. Lines 488-489

L 357: change was to were
The authors: corrected.

L 374: the word neat is confusing, please rephrase
The authors: corrected.

L 416: change supporting further to further support
The authors: corrected.

L 427: move “logically” to before “shows” or replace by “, as expected,”
The authors: corrected.

L 428: nicely is not appropriate
The authors: corrected.

L 487: add “that” before “are”
The authors: corrected.

L 489: delete “the”
The authors: corrected.

References cited

Adrain, J.M., 2013. Chapter 20: a synopsis of Ordovician trilobite distribution and diversity. In: Harper, D.A.T., Servais, T. (Eds.), *Early Palaeozoic Biogeography and Palaeogeography*. Geological Society, London, *Memoirs*, vol. 38. pp. 297–336. <https://doi.org/10.1144/M38.20>.

Cherns, L., Wheeley, J.R., Popov, L.E., Ghobadi Pour, M., Owens, R.M., Hemsley, A.R., 2013. Long-period orbital climate forcing in the early Palaeozoic? *Journal of the Geological Society*, London 170, 707-710.

Cole, S.R., 2018. Phylogeny and evolutionary history of diplobathrid crinoids (Echinodermata). *Palaeontology* 1–17. <https://doi.org/10.1111/pala.12401>.

Albanesi, G.L., and Bergström, S.M., 2010. Early–Middle Ordovician conodont paleobiogeography with special regard to the geographic origin of the Argentine Precordillera: A multivariate data analysis, in Finney, S.C., and Berry, W.B.N., eds., *The Ordovician Earth System: Geological Society of America Special Paper 466*, p. 119–139, doi: 10.1130/2010.2466(08).

Bergström, S.M. 1971. Conodont biostratigraphy of the Middle and Upper Ordovician of Europe and eastern North America. *Geological Society of America Memoir* 127, 83–157.

Bergström, S.M. 1986. Biostratigraphic integration of Ordovician graptolite and conodont zones—a regional review. In Hughes, C.P. & Rickards, R.B. (eds) *Palaeoecology and Biostratigraphy of Graptolites*. Geological Society of London Special Publication 30, 61–78.

Bergström, S.M. & Ferretti, A. 2017. Conodonts in Ordovician biostratigraphy. *Lethaia* 50, 424–439.

- Bergström, S.M. & Löfgren, A. 2009. The base of the global Dapingian Stage (Ordovician) in Baltoscandia: conodonts, graptolites and unconformities. *Earth and Environmental Transactions of the Royal Society of Edinburgh* 99, 189–212.
- Bergström, S.M., Chen, X., Gutiérrez-Marco, J.C. & Dronov, A. 2009. The new chronostratigraphic classification of the Ordovician System and its relations to major regional series and stages and to $\delta^{13}\text{C}$ chemostratigraphy. *Lethaia* 42, 97–107.
- Colmenar, J., Rasmussen, C.M.Ø., 2018. A Gondwanan perspective on the Ordovician Radiation constrains its temporal duration and suggests first wave of speciation, fuelled by Cambrian clades. *Lethaia* 51, 286–295.
- Dabard, M.P., Loi, A., Paris, F., Ghienne, J.F., Pistis, M., Vidal, M., 2015. Sea level curve for the Middle to early Late Ordovician in the Armorican Massif (western France): Icehouse third order glacio-eustatic cycles. *Palaeogeography Palaeoclimatology Palaeoecology* 436, 96–111.
- Ernst, A., 2018. Diversity dynamics of Ordovician bryozoa. *Lethaia* 51, 198–206. <https://doi.org/10.1111/let.12235>.
- Fan, J.-x., Shen, S.-z., Erwin, D.H., Sadler, P.M., MacLeod, N., Cheng, Q.-m., Hou, X.-d., Yang, J., Wang, X.-d., Wang, Y., Zhang, H., Chen, X., Li, G.-x., Zhang, Y.-c., Shi, Y.-k., Yuan, D.-x., Chen, Q., Zhang, L.-n., Li, C., Zhao, Y.-y., 2020. A high-resolution summary of Cambrian to Early Triassic marine invertebrate biodiversity. *Science* 367, 272–277.
- Fang, Q., Wu, H., Wang, X., Yang, T., Li, H., Zhang, S., 2019. An astronomically forced cooling event during the Middle Ordovician. *Global and Planetary Change* 173, 96–108.
- Harper, D.A.T., Rasmussen, C.M.Ø., Liljeroth, M., Blodgett, R.B., Candela, Y., Jin, J., Percival, I.G., Rong, J.-y., Villas, E., Zhan, R.-b., 2013. Biodiversity, biogeography and phylogeography of Ordovician rhyconelliform brachiopods. Geological Society, London, *Memoirs* 38, 127–144.
- Harper, D., Cascales-Miñana, B., & Servais, T. 2020. Early Palaeozoic diversifications and extinctions in the marine biosphere: A continuum of change. *Geological Magazine*, 157(1), 5–21. doi:10.1017/S0016756819001298
- Hints, L., Harper, D.A.T., Paškevičius, J., 2018. Diversity and biostratigraphic utility of Ordovician brachiopods in the East Baltic. *Est. J. Earth Sci.* 67, 176–191. <https://doi.org/10.3176/earth.2018.14>.
- V. Jaanusson, Aspects of carbonate sedimentation in the Ordovician of Baltoscandia. *Lethaia* 6, 11–34 (1973).
- Kröger, B., Franeck, F., Rasmussen, C.M.Ø., 2019. The evolutionary dynamics of the early Palaeozoic marine biodiversity accumulation. *Proceedings of the Royal Society B* 286, 20191634.
- Lindskog, M. E. Eriksson, Megascopic processes reflected in the microscopic realm: sedimentary and biotic dynamics of the Middle Ordovician “orthoceratite limestone” at Kinnekulle, Sweden. *GFF* 139, 163–183 (2017).
- Lindskog, A., Costa, M.M., Rasmussen, C.M.Ø., Connelly, J.N., Eriksson, M.E., 2017. Refined Ordovician timescale reveals no link between asteroid breakup and biodiversification. *Nature Communications* 8, 14066.
- Löfgren, A. 1978. Arenigian and Llanvirnian conodonts from Jämtland, Sweden. *Fossils and Strata* 13, 1–129.
- Rasmussen, C.M.Ø., Hansen, J., Harper, D.A.T., 2007. Baltica: A mid Ordovician diversity hotspot. *Historical Biology* 19, 255–261.
- Rasmussen, C.M.Ø., Nielsen, A.T., Harper, D.A.T., 2009. Ecostratigraphical interpretation of lower Middle Ordovician East Baltic sections based on brachiopods. *Geological Magazine* 146, 717–731.

- Rasmussen, C.M.Ø., Ullmann, C.V., Jakobsen, K.G., Lindskog, A., Hansen, J., Hansen, T., Eriksson, M.E., Dronov, A., Frei, R., Korte, C., Nielsen, A.T., Harper, D.A.T., 2016. Onset of main Phanerozoic marine radiation sparked by emerging Mid Ordovician icehouse. *Scientific Reports* 6, 18884.
- Rasmussen, C.M.Ø., Kröger, B., Nielsen, M.L., Colmenar, J., 2019. Cascading trend of early Paleozoic marine radiations paused by Late Ordovician mass extinctions. *PNAS* 116, 7207–7213.
- Rasmussen, J.A. 1998. A reinterpretation of the conodont Atlantic Realm in the late Early Ordovician (early Llanvirn). In: H. Szaniawski (ed.), *Proceedings of the Sixth European Conodont Symposium (ECOS VI)*. *Palaeontologia Polonica* 58, 67–77.
- Rasmussen, J.A. 2001. Conodont biostratigraphy and taxonomy of the Lower Ordovician shelf margin deposits in the Scandinavian Caledonides. *Fossils and Strata* 48, 1–180.
- Rasmussen, J.A. & Stouge, S. 2018. Baltoscandian conodont biofacies fluctuations and their link to Middle Ordovician (Darriwilian) global cooling. *Palaeontology* 61, 391–416, doi: 10.1111/pala.12348.
- Rong, J.Y., Fan, J.X., Miller, A.I., Li, G.X., 2007. Dynamic patterns of latest Proterozoic-Palaeozoic-early Mesozoic marine biodiversity in South China. *Geological Journal*. 42(3-4), 431-454.
- Saltzman, M.R., Young, S.A., 2005. Long-lived glaciation in the Late Ordovician? Isotopic and sequence-stratigraphic evidence from western Laurentia. *Geology* 33, 109-112.
- Schmitz, B., Harper, D.A.T., Peucker-Ehrenbrink, B., Stouge, S., Alwmark, C., Cronholm, A., Bergström, S.M., Tassinari, M., Xiaofeng, W., 2008. Asteroid breakup linked to the Great Ordovician Biodiversification Event. *Nature Geoscience* 1, 49-53.
- Schmitz, B., Farley, K.A., Goderis, S., Heck, P.R., Bergström, S.M., Boschi, S., Claeys, P., Debaille, V., Dronov, A., van Ginneken, M., Harper, D.A.T., Iqbal, F., Friberg, J., Liao, S., Martin, E., Meier, M.M.M., Peucker-Ehrenbrink, B., Soens, B., Wieler, R., Terfelt, F., 2019. An extraterrestrial trigger for the mid-Ordovician ice age: Dust from the breakup of the L-chondrite parent body. *Science Advances* 5.
- Sepkoski, J.J., Jr., 1979. A kinetic model of Phanerozoic taxonomic diversity. II. Early Phanerozoic families and multiple equilibria. *Paleobiology* 5, 222–251.
- Sepkoski, J.J., Jr., 1981. A factor analytic description of the Phanerozoic marine fossil record. *Paleobiology* 7, 36-53.
- Sepkoski Jr., J.J., 1995. The Ordovician Radiations: diversification and extinction shown by global genus level taxonomic data. In: Cooper, J.D., Droser, M.L., Finney, S.C. (Eds.), *Ordovician Odyssey: Short Papers, 7th International Symposium on the Ordovician System*, Pacific Section Society for Sedimentary Geology (SEPM), Book. vol. 77. Fullerton, California, pp. 393–396.
- Sepkoski, J.J., Jr., Sheehan, P.M., 1983. Diversification, Faunal Change, and Community Replacement during the Ordovician Radiations, In: Tevesz, M.J.S., McCall, P.L. (Eds.), *Biotic interactions in recent and fossil benthic communities*. Plenum Press, New York, pp. 673-717.
- Servais, T., Harper, D.A.T., 2018. The Great Ordovician Biodiversification Event (GOBE): definition, concept, and duration. *Lethaia* 51, 151–164. <https://doi.org/10.1111/let.12259>.
- Servais, T., Perrier, V., Danelian, T., Klug, C., Martin, R.E., Munnecke, A., Nowak, H., Nützel, A., Vandenbroucke, T., Williams, M., Rasmussen, C.M.Ø., 2016. The onset of the 'Ordovician Plankton Revolution' in the late Cambrian. *Palaeogeography Palaeoclimatology Palaeoecology* 458, 12–28.
- Servais, T., Cascales-Miñana, B., Harper, D.A.T.: The Great Ordovician Biodiversification Event (GOBE) is not a single event. *Paleontological Research*, doi: 10.2517/2021PR001.

Stigall, A.L., Edwards, C.T., Freeman, R.L., Rasmussen, C.M.Ø., 2019. Coordinated biotic and abiotic change during the Great Ordovician Biodiversification Event: Darriwilian Assembly of earliest Paleozoic building blocks. *Palaeogeography, Palaeoclimatology, Palaeoecology*.

Stouge, S., 1984, Conodonts of the Middle Ordovician Table Head Formation, Western Newfoundland: *Fossils and Strata* 16, 145 p.

Stouge, S., Bagnoli, G. & Rasmussen, J.A. 2020. Late Cambrian (Furongian) to mid-Ordovician euconodont events on Baltica: Invasions and immigrations. *Palaeogeography, Palaeoclimatology, Palaeoecology* 549, 16 pp. <https://doi.org/10.1016/j.palaeo.2019.04.007>

Trotter, J.A., Williams, I.S., Barnes, C.R., Lécuyer, C., Nicoll, R.S., 2008. Did cooling oceans trigger Ordovician biodiversification? Evidence from conodonts thermometry. *Science* 321, 550-554.

Trubovitz, S., Stigall, A.L., 2016. Synchronous diversification of Laurentian and Baltic rhynchonelliform brachiopods: Implications for regional versus global triggers of the Great Ordovician Biodiversification Event. *Geology* 44, 743-746.

Turner, B.R., Armstrong, H.A., Wilson, C.R., Makhlof, I.M., 2012. High frequency eustatic sea-level changes during the Middle to early Late Ordovician of southern Jordan: Indirect evidence for a Darriwilian Ice Age in Gondwana. *Sedimentary Geology* 251–252, 34–48.

Vandenbroucke, T.R.A., Armstrong, H.A., Williams, M., Paris, F., Zalasiewicz, J., Sabbe, K., Nölvak, J., Challands, T.J., Verniers, J., Servais, T., 2010. Polar front shift and atmospheric CO₂ during the glacial maximum of the Early Paleozoic Icehouse. *PNAS* 107, 14983–14986.

Webby, B.D., Paris, F., Droser, M.L., Percival, I.C., 2004. The Great Ordovician Biodiversification Event. Columbia University Press, New York, p. 484.

Zhan, R.-b., Harper, D.A.T., 2006. Biotic diachroneity during the Ordovician Radiation: evidence from South China. *Lethaia* 39, 211-226.

Zhang J. 1998. Conodonts from the Guniutan Formation (Llanvirnian) in Hubei and Hunan Provinces, south-central China. *Stockholm Contributions in Geology* 46: 1–161.

REVIEWERS' COMMENTS

Reviewer #1 (Remarks to the Author):

Dear authors and editor,

I understand the tone in the response to my comment is to avoid the rejection of this manuscript by NatureCommunications. However, the revised manuscript does address my concerns and mentions the diverse definitions of the GOBE, and clarifies their preferred usage of this term in the Discussion section. Moreover, the authors also acknowledge the work by Fang et al. (2019) and discuss the uncertainty of geochronology. They have tried their best to address all the comments. So, this manuscript is publishable.

Minor comment:

Fig. 6: orbital switch should be orbital shift (see the comment by reviewer #2).

Fig. 4 and fig. 7B: Use the same display style as used in fig. 5F.

Figs. 4-9. Too many colors exist in one figure.

Reviewer #2 (Remarks to the Author):

Dear authors,

Thank you for the revised version of your manuscript. The time series analyses results are presented clearly, and I particularly appreciate the close-up of the section with gray level variations which really gives the reader an impression of the style of the rhythmicity. The results from different time series analyses techniques are consistent and support the authors' interpretation.

I only have minor edits and recommend publication.

minor comments

l 12: consider adding 'and Mesozoic" after Cenozoic.

l 25: add "up" after leading

l 96: would it be possible to include a more detailed paleogeographic map with the terrestrial areas indicated?

l 211: change depicetd to "identified"

l 268: change provide to provides

l 269: change spline to "spline-fit"

l 553: I don't udnertsand "despite a significant alteration.." consider rephrasing.

l 589: change restitutes to "replicates" or another verb of choice

l 599: change are to "and"

Reviewer #3 (Remarks to the Author):

Dear Authors,

you have adequately addressed my concerns. Many thanks for your careful revisions.

Reading the other comments and your replies, I have two recommendations/comments:

Recommendation: You argue that the Study by Fang et al. 2019 is difficult to correlate. I agree with this, and the fact that their stratigraphy is not as refined as yours. Still, comparing your results in the manuscript would be a good idea in my opinion.

Comment: In your rebuttal you state that the study by Olsen et al. 2019 demonstrates the value of lithological rank systems. While I agree, please note that Olsen et al. also use colour and well logging data. In my opinion your demonstration of the proxy data relating to your lithology is suitable for publication.

Kind Regards,

REVIEWERS' COMMENTS

Reviewer #1 (Remarks to the Author):

Dear authors and editor,

I understand the tone in the response to my comment is to avoid the rejection of this manuscript by NatureCommunications. However, the revised manuscript does address my concerns and mentions the diverse definitions of the GOBE, and clarifies their preferred usage of this term in the Discussion section. Moreover, the authors also acknowledge the work by Fang et al. (2019) and discuss the uncertainty of geochronology. They have tried their best to address all the comments. So, this

manuscript is publishable.

Minor comment:

Fig. 6: orbital switch should be orbital shift (see the comment by reviewer #2).

Corrected

Fig. 4 and fig. 7B: Use the same display style as used in fig. 5F.

Corrected

Figs. 4-9. Too many colors exist in one figure.

We systematically replaced the green color by another (generally brown) so as not to mix green and red in all figures

Reviewer #2 (Remarks to the Author):

Dear authors,

Thank you for the revised version of your manuscript. The time series analyses results are presented clearly, and I particularly appreciate the close-up of the section with gray level variations which really gives the reader an impression of the style of the rhythmicity. The results from different time series analyses techniques are consistent and support the authors' interpretation.

I only have minor edits and recommend publication.

minor comments

l. 12: consider adding 'and Mesozoic" after Cenozoic.

Corrected

l. 25: add "up" after leading

Corrected

l. 96: would it be possible to include a more detailed paleogeographic map with the terrestrial areas indicated?

Unfortunately no, we cannot. The exact position of the land at that time is too loosely constrained.

l. 211: change depicted to "identified"

Corrected, we modified several "depicted" to avoid repetitions.

l. 268: change provide to provides

corrected

l. 269: change spline to "spline-fit"

corrected

l. 553: I don't understand "despite a significant alteration.." consider rephrasing.

We simply removed this part of the sentence which does not bring anything essential to the paper.

l. 589: change restitutes to "replicates" or another verb of choice

corrected

l. 599: change are to "and"

The reviewer meant the other way around. We modified "and showing" by "show" here so the sentence reads better

Reviewer #3 (Remarks to the Author):

Dear Authors,

you have adequately addressed my concerns. Many thanks for your careful revisions.

Reading the other comments and your replies, I have two recommendations/comments:

Recommendation: You argue that the Study by Fang et al. 2019 is difficult to correlate. I agree with this, and the fact that their stratigraphy is not as refined as yours. Still, comparing your results in the manuscript would be a good idea in my opinion.

Ok. We expanded this part of the discussion: “Our orbital shift from precession to 405 kyr eccentricity could possibly correspond to the conjunction of minima in the 1.2 Myr obliquity and 2.4 Myr eccentricity grand cycles demonstrated to take place in the Tarim Basin (NW China) at ca. 4.07 Myr prior to the Darriwilian/Sandbian boundary and claimed responsible for cooling and for the rise in diversity of graptoloids in this area¹⁷. The authors of the latter study found this conjunction of grand cycles minima within or below the North American *Histiodellella sinuosa* conodont Zone which correlates to a rather large interval of our study comprising the *L. antivariabilis* to base of *Y. crassus* Baltica conodont Zones . However, the latter authors suggest that this orbital event would postdate the base Darriwilian by ca. 2.1 Myr whereas our findings place our orbital shift within the first Myr postdating the base Darriwilian. Alternatively, our main orbital shift could also correspond to the 1.2 Myr obliquity minimum of ref. 17 that postdates the base Darriwilian by ca. 800 kyr and is in near-conjunction with a 2.4 Myr maximum. The comparison of our results to the study by Fang et al. remains tenuous due to the large uncertainty in the exact position of the base Darriwilian in both our study and theirs and in the correlation of our conodont zones to that of the Tarim Basin.” Lines 417-436.

Comment: In your rebuttal you state that the study by Olsen et al. 2019 demonstrates the value of lithological rank systems. While I agree, please note that Olsen et al. also use colour and well logging data. In my opinion your demonstration of the proxy data relating to your lithology is suitable for publication.

Kind Regards,